# Community assessment of methods to deconvolve cellular composition from bulk gene expression

Brian S. White [1,42], Aurélien de Reyniès[2], Aaron M. Newman [3,4], Joshua J. Waterfall [5], Andrew Lamb[1], Florent Petitprez [6,7], Yating Lin [8], Rongshan Yu [8], Martin E. Guerrero-Gimenez[9], Sergii Domanskyi[10], Gianni Monaco [11], Verena Chung [1], Jineta Banerjee [1], Daniel Derrick [12], Alberto Valdeolivas[13], Haojun Li[8], Xu Xiao [8], Shun Wang [14], Frank Zheng[15], Wenxian Yang [16], Carlos A. Catania[17], Benjamin J. Lang [18], Thomas J. Bertus[10], Carlo Piermarocchi[10], Francesca P. Caruso [11], Michele Ceccarelli [11,19], Thomas Yu[1], Xindi Guo[1], Julie Bletz[1], John Coller[20], Holden Maecker [21], Caroline Duault [21], Vida Shokoohi[20], Shailja Patel[22], Joanna E. Liliental[22], Stockard Simon[1], Tumor Deconvolution DREAM Challenge consortium*, Julio Saez-Rodriguez [13], Laura M. Heiser [12], Justin Guinney[1] & Andrew J. Gentles [4,23,24] ✉

We evaluate deconvolution methods, which infer levels of immune infiltration from bulk expression of tumor samples, through a community-wide DREAM Challenge. We assess six published and 22 community-contributed methods using in vitro and in silico transcriptional profiles of admixed cancer and healthy immune cells. Several published methods predict most cell types well, though they either were not trained to evaluate all functional CD8+ T cell states or do so with low accuracy. Several community-contributed methods address this gap, including a deep learning-based approach, whose strong performance establishes the applicability of this paradigm to deconvolution. Despite being developed largely using immune cells from healthy tissues, deconvolution methods predict levels of tumor-derived immune cells well. Our admixed and purified transcriptional profiles will be a valuable resource for developing deconvolution methods, including in response to common challenges we observe across methods, such as sensitive identification of functional CD4+ T cell states.

Tissues are comprised of multiple cell types that interact to confer diverse functions. Cells of the immune system are increasingly recognized for their critical function in both normal and diseased tissues, and many diseases have been linked to changes in the immune context of tissues, including cancer, Alzheimer's disease, arthritis and the SARS-CoV-2 virus, responsible for the COVID19 pandemic. In the field of oncology, the immune system has emerged as a critical factor in determining disease progression, patient survival, and response to therapy. The tumor microenvironment (TME) also presents a number of actionable therapeutic targets[1]. Notably, immune checkpoint inhibitors, which are designed to re-potentiate cytotoxic T cells to engage and kill malignant cells, have led to spectacular clinical outcomes for a subset of patients with previously dire prognoses. The precise reasons why some patients respond, but others do not, are

---

A full list of affiliations appears at the end of the paper. *A list of authors and their affiliations appears at the end of the paper. ✉e-mail: andrewg@stanford.edu

poorly understood, indicating that a more detailed understanding of the TME is needed.

Single-cell sequencing, imaging, and quantification platforms can directly characterize the TME[2,3], yet analysis of existing and new bulk tissue transcriptomic profiles remain significant for two principal reasons: (1) The community has invested heavily in large databases of clinically annotated samples with bulk data such as The Cancer Genome Atlas (TCGA) and the Gene Expression Omnibus (GEO). (2) Bulk measurements, in particular RNA-seq, continue to be regularly employed owing to challenges with more advanced single-cell platforms, including their cost and/or required specialized equipment [e.g., for CODEX[4], mass cytometry (CyTOF), fluorescence activated cell sorting (FACS), imaging mass cytometry (IMC)[5], multiplex ion beam imaging (MIBI)[6], and spatial transcriptomics (ST)[7]; difficulties in tissue sample preparation[8]; biases induced by preparation and/or single-cell dissociation, such as transcriptome-altering cellular stress and hypoxia[9,10]; and under-representation of specific cell types, such as fragile, short-lived neutrophils[11,12]. Hence, both retrospective and prospective correlative studies of immune infiltration with clinical outcome require dissecting bulk measurements into their constituent cell types and remain of great interest and value to the community.

Computational "deconvolution" methods meet this need by inferring the relative proportions of specific cell types from bulk RNA-seq or microarray transcriptional profiles[13–18]. Application of these have demonstrated, for example, that tumor B and CD8+ T cell proportions are predictive of immune checkpoint inhibitor response across different cancers[19–21], and how infiltration of diverse immune cell types variably impact patient prognosis[22]. Unsupervised deconvolution methods dissect cellular composition without a priori information about the cell types such as marker genes or expression profiles[23]. Supervised approaches instead consist of two main classes: reference-based deconvolution methods estimate the fractions of cell types in a mixture (or RNA contribution to the total RNA of the mixture), while enrichment-based methods assign a per-cell-type score that can be used to compare the relative prevalence of a specific cell type across samples, but cannot compare different cell types. Enrichment-based methods can sensitively distinguish between "coarse-grained" cell types (e.g., B versus T cells), but often have low specificity in discriminating between "fine-grained" cell types (e.g., sub-populations of T cells such as central memory CD4+ T cells, effector CD8+ T cells, or Tregs)[24]. Reference-based deconvolution methods are typically more specific than enrichment-based methods, but may be less sensitive. The tradeoff between these properties is important when considering their application to particular questions.

Several benchmarking efforts have used in silico simulation to evaluate factors that impact the accuracy of published deconvolution methods, including technical noise[25] and the specificity of marker genes[24]. The latter study by Sturm and colleagues assessed six published methods and reported that they robustly predict well-characterized, coarse-grained cell types (e.g., CD8+ T cells, B cells, NK cells, and fibroblasts), but were less accurate in predicting fine-grained sub-populations, including Tregs. These prior studies motivated our design of a community-wide DREAM Challenge to encourage development of new methods to deconvolve the cellular composition of fine-grained sub-populations from bulk gene expression and to rigorously benchmark these against published methods. DREAM Challenges (https://dreamchallenges.org/) are a well-established framework, and have benchmarked >60 bioinformatic algorithms, leveraging a community of >30,000 participants. We chose to focus on supervised methods, both reference- and enrichment-based, as these are most widely used in the cancer community. We use the overarching term "deconvolution" to refer to both reference- and enrichment-based methods.

To facilitate our benchmarking efforts, we generated in vitro admixtures for use as held-out validation data. We extracted RNA from cancer cells and healthy immune and stromal cells, combined them in proportions representative of solid tumors, and performed RNA-seq on them. Additionally, we generated in silico admixtures from the expression profiles of the same purified samples used in the in vitro admixtures. In both cases, the known mixing proportions were used as ground truth in assessing method predictions from the resulting bulk tissue expression using correlation as a metric. Simulating (in vitro or in silico) admixtures allowed us to define controlled ground truth in isolation from experimental biases due to technical and biological variability. We spurred development of 22 new methods contributed by participating international teams. Our resulting benchmark of these participant methods alongside six published deconvolution methods that are widely used by the community is the largest to date.

Consistent with Sturm and colleagues, we find that most methods predict coarse-grained populations well. Beyond that, several participant methods improved prediction of fine-grained populations, including memory and naïve CD8+ T cells, by leveraging the broad cell type coverage provided by our Challenge data. We generated in silico admixtures from single-cell RNA-seq (scRNA-seq) profiles of tumor samples and used them to show that these methods, trained largely on immune expression profiles from *healthy* tissues, also deconvolve *cancer-associated* immune cells well. Through its success in our Challenge, we demonstrate the applicability of deep learning to deconvolution as an alternative methodology to previously employed reference- and enrichment-based approaches. Though neither it nor any other single method performed best across all cell types, we showed that an ensemble approach combining all methods exploits their individual strengths. The scope of our study allowed us to detect pervasive difficulties in the field shared across a wide range of methods, diverse in their computational core and the manner in which they were trained, including challenges in sensitively predicting CD4+ T cell functional states. These results should contribute to greater robustness of deconvolution-based analyses, as they will enable researchers to select a method appropriate for deconvolving a particular cell type, or, alternatively, to appreciate the limitations of these methods in doing so.

The expression profiles of purified populations and in vitro admixtures generated for this Challenge are available as a resource for developing and training deconvolution methods in contexts where quantifying immune and stromal cells is of interest. Specifically, they can be used to develop methods addressing common shortcomings revealed by our Challenge (e.g., with regards to detecting CD4+ T cell functional states). Further, existing methods can be re-trained against this standard reference profile so as to evaluate them solely according to their core computational algorithms independent of differences arising from choice of training datasets.

## Results

### Purified and admixed expression profiles enable unbiased assessment of deconvolution methods

Immune cell infiltration has prognostic significance across multiple levels of immune cell specialization and polarization. For example, CD8+ T cells, broadly encompassing memory and naïve compartments, have positive prognostic value in many cancer types, whereas regulatory T cells, a specific subset of CD4+ T cells, generally have negative prognostic associations[26]. To assess deconvolution across these levels, we divided the Tumor Deconvolution DREAM Challenge into coarse-grained and fine-grained sub-Challenges. In the coarse-grained sub-Challenge, participants predicted levels of eight major immune and stromal cell populations: B cells, CD4+ and CD8+ T cells, NK cells, neutrophils, cells of monocytic lineage (monocytes, macrophages, and dendritic cells), endothelial cells, and fibroblasts. The fine-grained sub-Challenge further dissected these populations into 14 sub-populations, e.g., memory, naïve, and regulatory CD4+ T cells (Fig. 1A).

To facilitate benchmarking and create a ground truth dataset, we generated in vitro and in silico expression profiles of cell populations

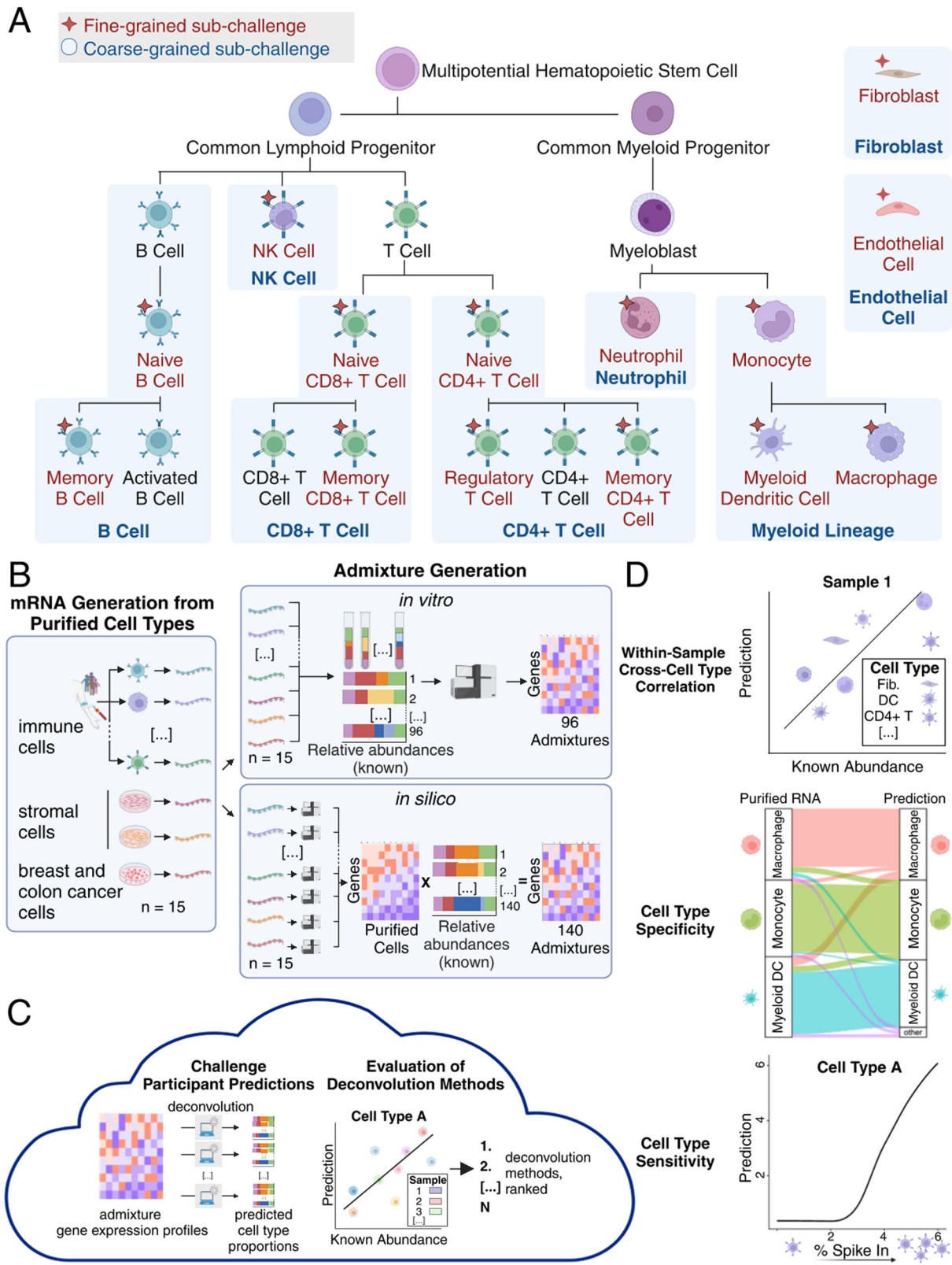

admixed at predefined ratios. We isolated immune cells from healthy donors and obtained stromal, endothelial, and cancer cells from cell lines (Fig. 1B and Supplementary Data 1 and 2; Methods). Cell type-specific marker expression was confirmed on the purified cells through RNA sequencing (Fig. S1). To robustly test algorithm performance across diverse conditions, we defined mixing proportions and

generated admixtures from them, grouped into one of eight datasets according to whether they: (1) included breast (BRCA) or colon cancer (CRC) cells; (2) had proportions that were unconstrained or constrained by biologically reasonable expectations ("biological" distribution); or (3) were created in silico or in vitro (Methods; Supplementary Data 3–8). This resulted in a total of 96 in vitro

**Fig. 1 | Generation of in silico and in vitro admixtures of immune, stromal, and cancer cells and their use as validation data for a DREAM Challenge. A** Cell populations predicted within fine-grained and coarse-grained sub-Challenges indicated with red text and star or blue text and blue shading, respectively. Cell types aggregated together in coarse-grained sub-Challenge are connected via their blue shading (e.g., monocytes, myeloid dendritic cells, and macrophages were classified as monocytic lineage). Immune populations are depicted within the haematopoietic hierarchy, which represents differential trajectories and not necessarily levels of specificity. **B** Admixture generation and use for validation. (Left) Purified immune cell populations were obtained from vendors and volunteers. Purified stromal and cancer cell populations were obtained from cell lines. (Right) In vitro admixtures were created by mixing mRNA from purified cell populations in specified ratios (unconstrained or biologically reasonable) and then subjected to RNA-seq. In silico admixtures were created by first sequencing purified cells to define population-specific signatures and then taking a linear combination of those signatures using specified ratios (unconstrained or constrained according to biologically reasonable expectation). **C** Deconvolution methods executed in the cloud against in silico and in vitro admixtures yielded predictions that were then compared to the input ratios using cross-sample, within-cell type correlation (Figs. 2 and 3). Methods were ranked according to their cross-sample, within-cell type Pearson correlations (primary metric), with ties resolved using cross-sample, within-cell type Spearman correlations. **D** Method performance was also quantified according to cross-cell type, within-sample correlation (Fig. 4), specificity (i.e., the spillover prediction from a purified cell type into a different cell type; Fig. 5), and sensitivity (i.e., the limit of detection for a particular cell type; Fig. 6). **A–D** Created with BioRender.com, released under a Creative Commons Attribution-NonCommercial-NoDerivs 4.0 International license.

admixtures and 140 in silico admixtures, with at least 18 admixtures in each dataset to ensure adequate sample size (Table S1). We generated in silico admixtures as a linear combination of the mixing proportions and the purified expression profiles and in vitro admixtures by extracting RNA from the purified cells, mixing them at the specified proportions, and sequencing (Fig. 1B).

We provided participants with a curated set of publicly available transcriptional profiles of purified cell types to use for training (Supplementary Data 9 and 10). Importantly, these training samples were not used in generating our own admixtures. Methods were evaluated against the admixtures by correlating the predictions of cell type proportions with the predefined (ground truth) proportions, independently for each cell type. Because the Challenge focused on microenvironmental populations, we only assessed participants' predictions based on inference of immune and stromal cell content, with the admixed cancer cells effectively treated as contaminating noise. We ranked methods with an aggregate score that averaged correlations across cell types and validation datasets (Fig. 1C; Methods). Methods were first assessed using a primary Pearson correlation-based score. Statistical ties were resolved relative to the top-performing method (as determined by a Bayes factor; Methods) by a secondary Spearman correlation-based score. To account for sampling variability, we reported these Pearson-based ($r$) and Spearman-based ($\rho$) scores as means across bootstraps (Methods). Following the conclusion of the ranked competition, we further quantified methods based on their cross-sample, within-cell type correlation, their specificity (i.e., the "spillover" of their prediction from one cell type into a different one), and their sensitivity (i.e., their limit of detection of a particular cell type; Fig. 1D).

As in previous DREAM Challenges[27], participants submitted their methods as Docker containers, which were executed in the cloud against held-out validation data. This "model-to-data"[28] approach ensured that data were not leaked to participants, prevented model over-fitting, and facilitated Challenge reproducibility. Twenty-two teams contributed 39 submissions (i.e., unique methods) to the coarse-grained sub-Challenge, while 16 teams contributed 48 submissions to the fine-grained sub-Challenge. Additionally, we applied six widely used published tools (CIBERSORT[13], CIBERSORTx[29], EPIC[14], MCP-counter[15], quanTIseq[17], and xCell[18]) as baseline comparator methods (Tables 1–2 and Methods section Comparator method description). Briefly, CIBERSORT[13] computes the linear combination of cell type expression profiles that optimally approximates the observed admixture expression over a set of markers, using $\nu$-support vector regression ($\nu$-SVR). CIBERSORTx[29] uses the same, $\nu$-SVR computational core as CIBERSORT, but additionally corrects for differences between reference and input admixture data. EPIC[14] computes the linear combination of cell-type expression profiles that optimally approximates the observed admixture expression over a set of markers, using constrained, weighted, least-squares optimization. MCP-counter[15] computes a cell type enrichment score as the arithmetic mean of the expression of that cell type's markers in linear expression space.

quanTIseq[17] computes the linear combination of cell-type expression profiles that optimally approximates the observed admixture expression over a set of markers, using constrained least-squares optimization. xCell[18] computes a cell type score using single-sample gene set enrichment analysis (ssGSEA) of the cell type's markers that it then calibrates to a linear scale. Comparator methods were run by Sage Bionetworks independent of method developers to ensure unbiased assessments, using default parameters and published reference signatures. No optimization was applied to tune their performance.

### Diverse deconvolution algorithmic cores perform well

We tested the performance of the six comparator methods, as well as Challenge participant methods. The median Pearson correlation-based score across participant and comparator methods was 0.75 [interquartile range (IQR): 0.43–0.81; Fig. 2A] for the coarse-grained sub-Challenge and 0.61 (IQR: 0.53–0.62; Fig. 2B) for the fine-grained sub-Challenge. Methods differ in their output. Some produce an arbitrary score proportional to the presence of a cell type, which can be used to compare the same cell type across samples, but not across cell types. Others generate normalized scores, non-negative proportions summing to one, or non-negative fractions that need not sum to one, which can be compared both across samples and cell types. Most approaches were computationally efficient, with a median execution time across methods for the 236 validation samples of 132 and 156 s for the coarse- and fine-grained sub-Challenges, respectively (Table 2 and Supplementary Data 11).

Across both comparator and participant approaches, CIBERSORTx was the top-performing method in the coarse-grained sub-Challenge according to both metrics ($r = 0.90$; $\rho = 0.83$; Fig. 2A). The next highest-scoring method and the top-performing participant method according to the primary Pearson-based score was Aginome-XMU ($r = 0.85$; ref. 30). Aginome-XMU was the top-performing method (participant or comparator) in the fine-grained sub-Challenge ($r = 0.76$; $\rho = 0.64$; Fig. 2B). Methods whose published reference signatures do not include all cell types in a sub-Challenge (e.g., five of six comparator methods, CIBERSORT, CIBERSORTx, EPIC, MCP-counter, and quanTIseq, in the fine-grained sub-Challenge) were not considered in that sub-Challenge's aggregate ranking. Additionally, there was broad consistency in method ranking across the two sub-Challenges, with the three top-ranked participant methods in the coarse-grained sub-Challenge (Aginome-XMU, DA_505, and Biogem) amongst the top seven evaluable methods in the fine-grained sub-Challenge. Conversely, the three top-ranked evaluable teams in the fine-grained sub-Challenge (Aginome-XMU, mitten_TDC19, and DA_505) were amongst the top seven in the coarse-grained sub-Challenge.

We also compared the performance of deconvolution methods to an ensemble combination of their outputs as a potential upper bound. We defined an ensemble prediction as the consensus rank across individual methods and found that overall it outperformed any

**Table 1 | Deconvolution approach and marker selection for comparator and top-performing participant methods**

| | Objective Function | Type | Solver | Marker Selection | Value | Ref |
|---|---|---|---|---|---|---|
| CIBERSORT | $\operatorname{argmin}_{\beta,\nu,\epsilon} C \sum_m \lvert a_m - \mathbf{s}_m \cdot \beta \rvert_\epsilon + \frac{1}{2}\lVert\beta\rVert^2 + \nu\epsilon$ | Reference | svm | Differentially expressed (DE'ed), prioritized by fold change, not expressed in non-hematopoetic cells | $\beta^+$ | 13 |
| CIBERSORTx | $\operatorname{argmin}_{\beta,\nu,\epsilon} C \sum_m \lvert a_m^* - \mathbf{s}_m \cdot \beta \rvert_\epsilon + \frac{1}{2}\lVert\beta\rVert^2 + \nu\epsilon$ | Reference | svm | DE'ed, prioritized by fold change, not expressed in non-hematopoetic cells | $\beta^+$ | 29 |
| EPIC | $\operatorname{argmin}_{\beta}$ $\sum_m w_m^{\text{EPIC}}(a_m - \mathbf{s}_m \cdot \beta)^2$ s.t. $\beta_m \geq 0\ \forall m$ $\sum_m \beta_m \leq 1$ | Reference | constr-Optim | DE'ed, not expressed in non-hematopoetic tissues, similarly expressed in healthy and malignant tissues | $\beta$ | 14 |
| MCP-Counter | | Enrichment | | DE'ed across hierarchy of purified expression profiles, specific to cell type | $\frac{1}{\lvert M_c \rvert}\sum_{m \in M_c} a_m$ | 15 |
| quanTIseq | $\operatorname{argmin}_{\beta}$ $\sum_m (a_m - \mathbf{s}_m \cdot \beta)^2$ s.t. $\beta_m \geq 0\ \forall m$ $\sum_m \beta_m \leq 1$ | Reference | lsei | Correlated with random fraction in simulated admixtures, specific to cell type, expressed in tumors, not expressed in non-hematopoetic tissues, not very highly expressed | $\beta$ | 17 |
| xCell | | Enrichment | | DE'ed, specific to cell type, not expressed in carcinomas | ssGSEA mapped to linear scale | 18 |
| Aginome-XMU | DNN trained to predict random fractions in simulated admixtures | Other | | None | Fractions predicted by DNN | 30 |
| Biogem | $\operatorname{argmin}_{\beta} \sum_m [w_H(a_m - \mathbf{s}_m \cdot \beta)]^2 (a_m - \mathbf{s}_m \cdot \beta)^2$ | Reference | rlm | DE'ed, prioritized by effect size, not very highly or lowly expressed | $\beta$ | 33 |
| DA_505 | $\operatorname{argmin}_{\mathbf{b_c}} \lVert \widetilde{\mathbf{p}}_c - \widetilde{\mathbf{A}}_{M_c} \cdot \mathbf{b}_c \rVert^2 + \lambda_2 \lVert \mathbf{b}_c \rVert^2 + \lambda_1 \lVert \mathbf{b}_c \rVert_1$ | Other | | Identified by RF regression against random fraction in simulated admixtures | $\mathbf{A}_{M_c} \cdot \mathbf{b}_c$ | |
| mitten_TDC19 | | Enrichment | | Correlated with random fraction in simulated admixtures | $\sum_{m \in M_c} a_m$ | |

$M_c$: set of markers for cell type $c$.

$\mathbf{A}_{M_c}$, $\widetilde{\mathbf{A}}_{M_c}$: input or simulated admixture matrices, respectively, subset to markers for cell type $c$.

$\mathbf{a}$, $\mathbf{a}^*$: input or batch-corrected admixture expression vector, respectively.

$a_m$, $a_m^*$: expression for marker $m$ in input admixture $\mathbf{a}$ or batch-corrected admixture $\mathbf{a}^*$

$\mathbf{S}$: marker x cell type signature matrix.

$\mathbf{s}_m$: expression vector for marker $m$ across cell types (i.e., column of $\mathbf{S}$).

$\beta^+ \equiv (\beta_0^+, \beta_1^+, \ldots)$ with $\beta_i^+ \equiv \max(\beta_i, 0)$.

$\lvert e \rvert_\epsilon \equiv 0$ if $\lvert e \rvert < \epsilon$; $\lvert e \rvert - \epsilon$ otherwise.

$w_H(e) \equiv 1$ if $\lvert e \rvert < k$; $k/\lvert e \rvert$ otherwise.

$w_m^{\text{EPIC}}$: weight giving marker $m$ importance relative to its variability.

$\widetilde{\mathbf{p}}_c$: vector of proportions of cell type $c$ in simulated admixtures.

individual method in both the coarse- ($\rho = 0.84$; Fig. 2A) and fine-grained ($\rho = 0.67$; Fig. 2B) sub-Challenges. However, there was only modest improvement of the consensus rank method relative to the top-scoring individual methods by Spearman correlation in the coarse- (CIBERSORTx $\rho = 0.83$) and fine-grained (Aginome-XMU $\rho = 0.64$) sub-Challenges, suggesting that individual methods are not leveraging independent or orthogonal signals in the data despite their diverse approaches.

Several core algorithmic approaches were common across submissions, including those based on non-negative least squares (NNLS; 6 in the coarse-grained and 6 in the fine-grained sub-Challenge, respectively; Fig. 2A, B; Tables 1, 2) and summarization (SUM; 5 and 4). Nevertheless, there was wide methodological diversity amongst top performers. CIBERSORTx uses ν-SVR to simultaneously solve for all fractional abundances relating admixed and purified expression profiles using a signature matrix of ~525 differentially expressed genes spanning 22 immune cells types (LM22)[13,29]. Aginome-XMU, published subsequent to the Challenge[30], utilizes a neural network composed of an input layer, five fully connected hidden layers, and an output layer (Supplementary Methods; Tables S2 and S3; https://github.com/

xmuyulab/DCTD_Team_Aginome-XMU; Supplementary Data 12). The network effectively applies feature selection automatically and was trained here using synthetic admixtures. DA_505 applies a rank-based normalization, selects features by applying random forests to synthetic admixtures, and ultimately applies regression to predict abundance of each cell type *independently* (Supplementary Methods; https://github.com/martinguerrero89/Dream_Deconv_Challenge_Team_DA505; Fig. S2; Supplementary Data 13). mitten_TDC19 calculates a summarization score as the sum of the expression of selected markers, with the cell type-specific markers first nominated from expression profiles in purified bulk data or identified[31,32] from single-cell data expression profiles and then prioritized according to their correlation with that cell type's proportion over synthetic admixtures (Supplementary Methods; https://github.com/sdomanskyi/mitten_TDC19; Supplementary Data 14). Finally, Biogem, based on a previously published method[33], uses robust linear modeling to perform deconvolution and differential expression-based feature selection to define the purified expression profiles (Supplementary Methods; Tables S4–S6; https://github.com/giannimonaco/DREAMChallenge_Deconvolution; Fig. S3; Supplementary Data 15). Hence, despite their

**Table 2 | Training set, normalization, batch correction, and run times of comparator and top-performing participant methods**

| | Training Set | Normalization | Training Batch Correction | Test Batch Correction | Coarse Run-time (s) | Fine Run-time (s) |
|---|---|---|---|---|---|---|
| CIBERSORT | Healthy | Quantile | Same platform (HGU133A) | | 133 | 131 |
| CIBESORTx | Healthy (immune subtypes), Malignant (tumor/immune/stroma) | Quantile (immune only) | One platform for immune (HGU133A), One dataset for tumor/immune/stroma | Re-optimizes β w.r.t. a ComBat-corrected admixture matrix | 567 | 567 |
| EPIC | Healthy | TPM | | Re-normalizes TPMs based on genes common to **S** and test admixture | 71 | 70 |
| MCP-Counter | Healthy | fRMA | | NA | 70 | 66 |
| quanTIseq | Healthy | TPM | | | 253 | 254 |
| xCell | Healthy | FPKM (RNA-seq), RMA (microarray) | Markers determined independently in each training dataset | | 70 | 74 |
| Aginome-XMU | Healthy, Malignant | TPM, log, min-max scaling | | | 80 | 116 |
| Biogem | Healthy, Malignant | TPM | | | 82 | 81 |
| DA_505 | Healthy, Malignant | Rank-based | | | 92 | 90 |
| mitten_TDC19 | Healthy | TPM | | NA | 86 | 88 |

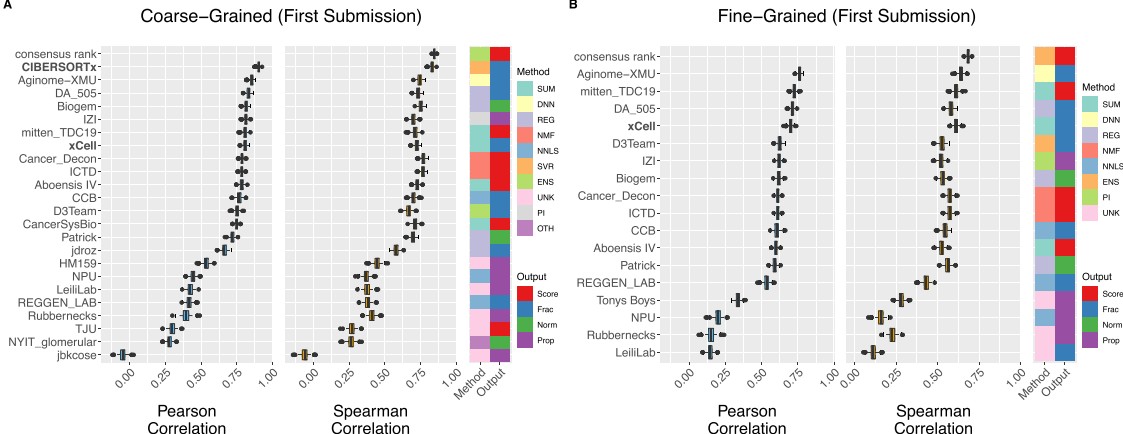

**Fig. 2 | Aggregate cross-sample performance of participant and comparator deconvolution methods.** Aggregate score (primary metric: Pearson correlation; secondary metric: Spearman correlation) of participant (first submission only) and comparator methods in (**A**) coarse- and (**B**) fine-grained sub-Challenges over bootstraps ($n = 1000$; Methods). Comparator methods (bold) are shown only if their published reference signatures include all cell types in each respective sub-Challenge: CIBERSORTx (coarse-grained only) and xCell. Boxplots display median (center line), 25th and 75th percentiles (hinges), and 1.5x interquartile range (whiskers). Methods ordered by median Pearson correlation in respective sub-Challenge. DNN deep neural network, ENS ensemble, NMF non-negative matrix factorization, NNLS non-negative least squares, OTH other, PI probabilistic inference, REG other regression, SUM summary, SVR support vector regression, UNK unknown/unspecified, Frac unnormalized fractions that need not sum to one, Norm normalized scores (comparable across cell types and samples), Prop proportions that sum to one. Source data are provided as a Source Data file.

algorithmic differences, three of the top-performing methods were trained using synthetic admixtures, generated in silico from publicly available purified expression profiles (Table S7). Importantly, the purified profiles that we created to generate Challenge admixtures were not made available to participants.

Method performance improved for most teams over three allowed submissions (Fig. S4), as they were permitted to revise their method with each submission. Since we provided teams with both aggregate and per-cell type scores following each submission, we can not exclude the possibility that these were used to tune, and potentially over-fit, methods between submissions. As such, we focused our analyses on the first submission, unless otherwise stated (Fig. S5).

### Deconvolution performance differs by cell type
We assessed methods in their ability to predict individual cell type levels within an admixture (Fig. 3 and Figs S6–S15) and found that all

major lineage cell types could be predicted robustly by at least one method (max row of Fig. 3A and Figs S7, S9, S11, and S14). For example, though CD4+ T cell levels were most difficult to predict on average, even these could be predicted with a Pearson correlation of 0.86 by CIBERSORTx and xCell. Neutrophil levels were predicted most accurately, with 18 of 28 methods having a Pearson correlation of at least 0.90.

Baseline comparator methods, other than xCell, were not trained to predict all the fine-grained immune subtypes used in the Challenge. For example, none of quanTIseq, MCP-counter, or EPIC differentiate between memory and naïve CD4+ T cells; and only xCell differentiates between memory and naïve CD8+ T cells, though with low accuracy for both (Fig. 3; $r \leq 0.40$). Participant models showed potential at predicting these poorly covered cell types. For example, mitten_TDC19 improved upon comparator performance in predicting both naïve CD8+ T cells ($r = 0.90$ vs xCell $r = 0.15$) and memory CD8+ T cells ($r = 0.82$ vs xCell $r = 0.40$), with Aboensis IV outperforming both for

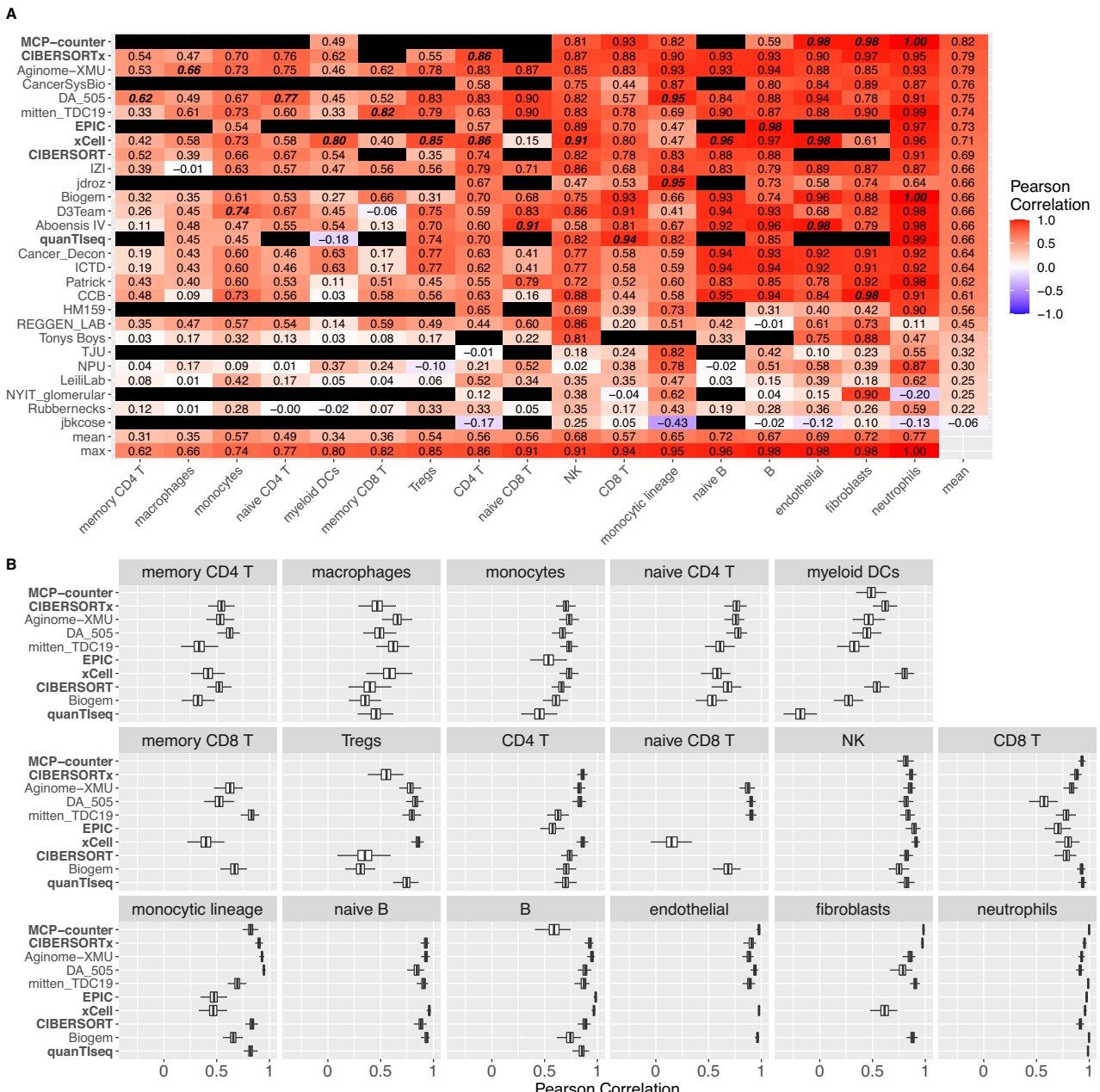

**Fig. 3 | Per-cell type performance of participant and comparator deconvolution methods. A** Pearson correlation of method (left axis) prediction versus known proportion from admixture for each cell type (bottom axis). Pearson correlation is first averaged over validation dataset and then over bootstraps ($n = 1000$; Methods) and subsequently averaged over coarse- and fine-grained sub-Challenges for cell types occurring in both. Black entry indicates cell type not predicted by corresponding method. Bottom two rows are the mean and maximum correlation, respectively, for corresponding cell type across methods. Rightmost column is mean correlation for corresponding method across predicted cell types. Highest correlations for each cell type highlighted in bold italics. **B** Performance (Pearson correlation; x axis) of comparator baseline methods and participant methods ranking within the top three in either or both sub-Challenges (y axis) for each cell type (facet label). Distribution of Pearson correlations over bootstraps ($n = 1000$; Methods), computed as average over validation datasets and subsequently over coarse- and fine-grained sub-Challenges for cell types occurring in both. Blank row indicates cell type not reported by the corresponding method. Comparator methods in bold. Boxplots display median (center line), 25th and 75th percentiles (hinges), and 1.5x interquartile range (whiskers). In both panels, methods ordered according to their mean Pearson correlation across cell types (rightmost column in (**A**)), and cell types ordered according to their maximum Pearson correlation across methods (bottom row in (**A**)). Source data are provided as a Source Data file.

naïve CD8+ T cells ($r = 0.91$). Further, Aginome-XMU performance on macrophages ($r = 0.66$ vs xCell $r = 0.58$) and DA_505 performance on memory CD4+ T cells ($r = 0.62$ vs CIBERSORTx $r = 0.54$) improved upon their respective best-performing comparator methods. In all other cases, participant methods showed some modest improvement (change in $r < 0.05$) relative to comparator methods. Notwithstanding these advances, the seven most difficult populations to predict were

functional subsets of CD4+ and CD8+ T cells and sub-populations of the monocytic lineage (Fig. 3).

The methods that performed well in aggregate also performed well based on individual cell types, though none dominated across all populations. Nominally, 10 methods were the top performers across one or more of the 17 individual cell populations. For most of these populations, multiple methods achieved similar performance to the

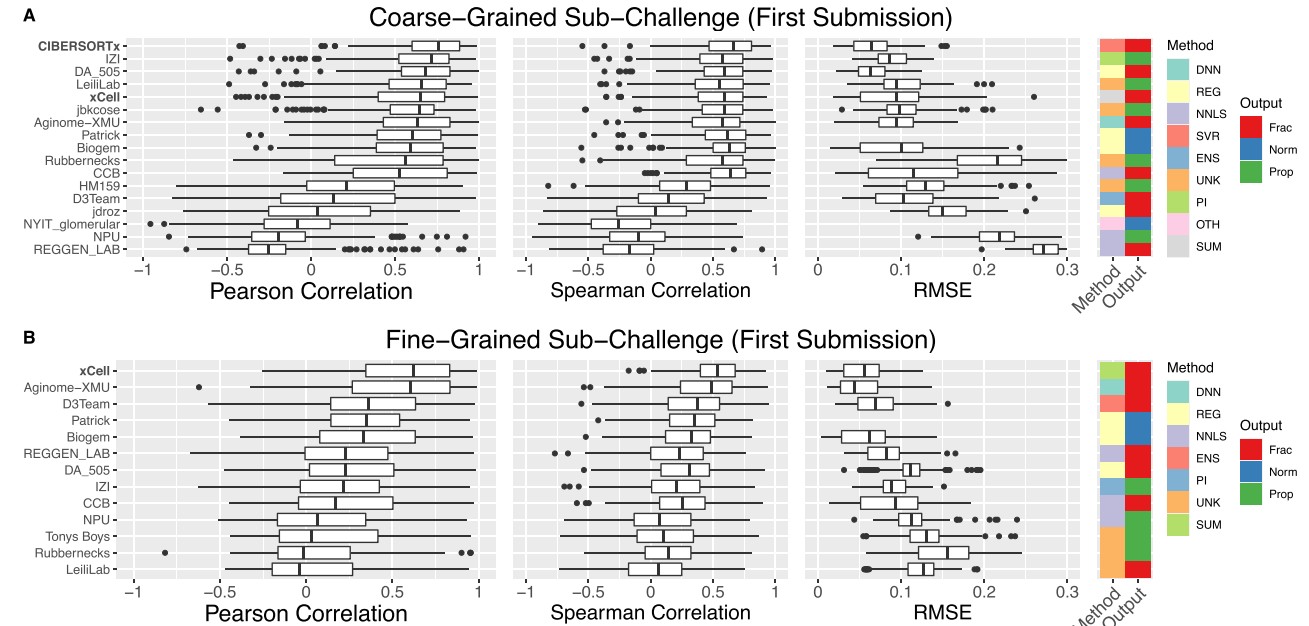

**Fig. 4 | Aggregate cross-cell type performance of participant and comparator deconvolution methods.** Performance [Pearson correlation, Spearman correlation, and root mean square error (RMSE)] of methods capable of within-sample, cross-cell type comparison to ground truth proportions in (**A**) coarse- and (**B**) fine-grained sub-Challenges. Distribution over $n = 166$ samples (methods ordered by median Pearson correlation across samples in respective sub-Challenge). Comparator methods in bold. Boxplots display median (center line), 25th and 75th percentiles (hinges), and 1.5x interquartile range (whiskers). DNN deep neural network, ENS ensemble, NNLS non-negative least squares, OTH other, PI probabilistic inference, REG other regression, SUM summary, SVR support vector regression, UNK unknown/unspecified, Frac fraction, Norm normalized score (comparable across cell types and samples), Prop proportion. Source data are provided as a Source Data file.

top-ranked one (Fig. 3B and Figs S6, S8, S10, S12, S13, and S15). Exceptions in which the top method outperformed the nearest method (comparator or participant) by a large margin (change in $r > 0.05$) were: xCell on myeloid dendritic cells ($r = 0.80$ vs Cancer_Decon $r = 0.63$), mitten_TDC19 on memory CD8+ T cells ($r = 0.82$ vs Biogem $r = 0.66$), and DA_505 on memory CD4+ T cells ($r = 0.62$ vs CIBERSORTX $r = 0.54$). In all other cases, the top-performing method showed at best a marginal improvement (change in $r < 0.05$) relative to the next best-performing method (comparator or participant).

## Cross-cell type, within-sample deconvolution performance is worse than cross-sample, within-cell type performance

We next assessed prediction performance across cell types within samples for those methods that produced normalized scores, proportions, and fractions (Fig. 4). To do so, we computed a correlation (Pearson and Spearman) and the root-mean-square error (RMSE) across cell types within a sample and then reported the median of these respective values across samples. Top-performing methods varied across sub-Challenge and metric (Table S8), though several methods performed well in both the above aggregate cross-sample/within-cell type comparison and in this cross cell-type/within-sample comparison: CIBERSORTx was amongst the top performers (i.e., having the highest score or showing no statistical difference from the method with the highest score) across all metrics in the coarse-grained sub-Challenge; DA_505 was a top performer based on RMSE and Spearman correlation in the coarse-grained sub-Challenge; Biogem was a top performer based on Spearman correlation in the coarse-grained sub-Challenge; and Aginome-XMU was a top performer based on RMSE and Pearson correlation in the fine-grained sub-Challenge. Additionally, several other methods were amongst the top performers across one or more metrics in one or both sub-Challenges, including: (1) xCell, which computes a score for each cell type by applying single sample gene set enrichment analysis (ssGEA) to a set

of marker genes, transforms the scores to proportions using a calibration function, and finally compensates for spillover between similar cell types; (2) CCB, which extends the published NNLS-based EPIC method by applying ssGSEA to those populations not treated by EPIC and by relating those ssGSEA scores to proportions via a calibration function; and (3) Patrick, which uses excludes tumor-associated genes from the immune and stromal reference signatures and then performs constrained optimization in logarithmic space.

## Deconvolution specificity is lower for T cells than for other cell types

Methods sometimes attribute signal from one cell type to a different cell type, particularly for highly similar cell types such as sub-populations of CD4 T cells. To assess specificity, we quantified the "spillover" between cell types as a method's prediction for a particular cell type $X$ within a sample purified for cell type $Y \neq X$ (Fig. 5A and Figs S16–S17). Based on median spillover, methods had greatest specificity for neutrophils. Expectedly, methods had greater specificity for the coarse- relative to the fine-grained populations (Fig. 5B): the second largest increase in median spillover separates a group enriched in major cell types [neutrophils, NK cells, naïve and parental (naïve and memory) B cells, endothelial cells, monocytes/monocytic lineage cells, and fibroblasts] from a group enriched in minor cell types (macrophages, memory/naïve/regulatory/parental CD4+ T cells, memory/naïve/parental CD8+ T cells, and myeloid dendritic cells). The single largest increase in median spillover separates memory CD4+ T cells from the remaining cell types. Across cell types, CCB had the lowest (median) spillover in both the coarse- (Fig. 5C) and fine-grained (Fig. 5D) sub-Challenges. In both cases, it was followed by Aboensis IV, a summarization-based approach that defined robust marker genes within a cell type mutually correlated with one another. The top-performing methods (Aginome-XMU, Biogem, CIBERSORTx, DA_505,

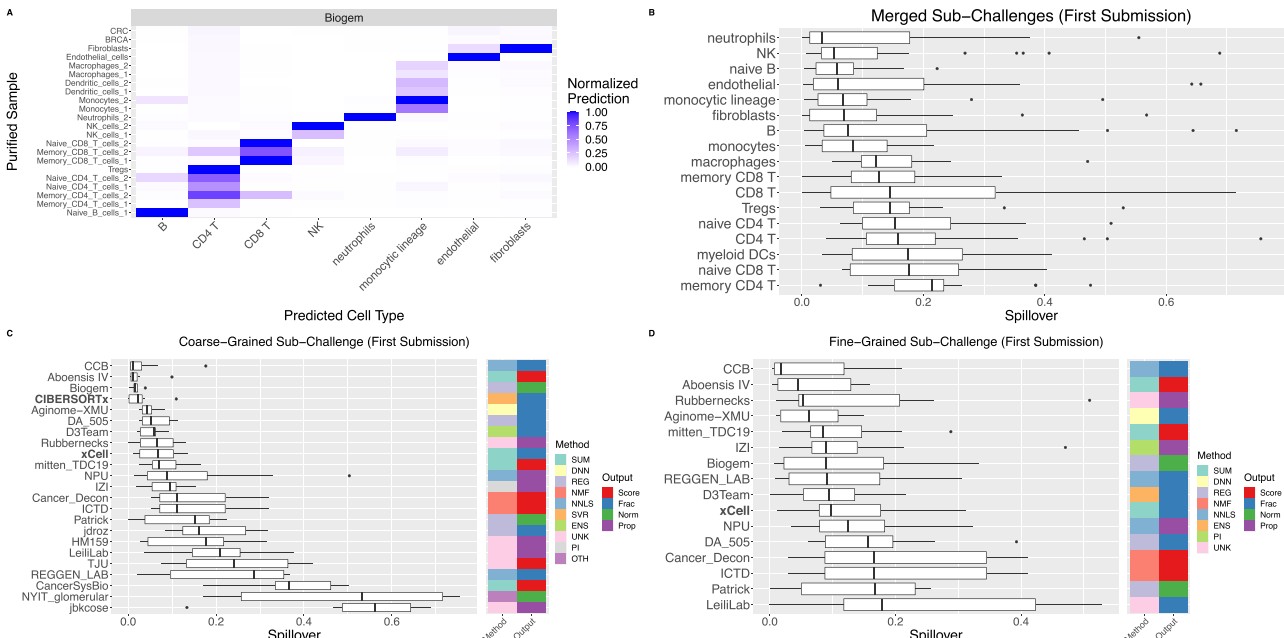

**Fig. 5 | Specificity of participant and comparator deconvolution methods.**
**A** Normalized prediction of cell type indicated on x axis in purified sample indicated on y axis. **B** Distribution over methods of spillover into cell type indicated on y axis (averaged first over samples purified for any other cell type, then over sub-Challenges; Methods). Cell types ordered according to their median spillover.

Distribution over cell types of spillover for each method in (**C**) coarse- and (**D**) fine-grained sub-Challenges. Methods ordered according to their median spillover. Comparator methods in bold. Boxplots display median (center line), 25th and 75th percentiles (hinges), and 1.5x interquartile range (whiskers). Source data are provided as a Source Data file.

and mitten_TDC19) rank within the top half of methods in both sub-Challenges.

## Deconvolution sensitivity is lower for CD4+ T cells than for other cell types

In real tumors, the representation of different cell types can range from only a fraction of a percent to a large proportion of the tissue. The limit of detection by deconvolution is likely to vary from cell type to cell type dependent on the uniqueness and strength of their transcriptional signal. We assessed deconvolution sensitivity using in silico spike in experiments (Methods). We spiked each cell type at a given frequency (ranging from 0% to 40%) into an unconstrained admixture of all other cell types. We then determined the minimum frequency at and above which that cell type could be distinguished from the baseline (0% spike in; Fig. 6A). The lowest limit of detection for any cell type was <0.2% (CIBERSORTx and Patrick for neutrophils and MCP-counter for CD8+ T cells), similar to that observed in prior microarray studies[15]. Seven methods showed similar mean limits of detection (3–4%) within the coarse-grained sub-Challenge (Fig. 6B), including top-ranked methods Biogem, CIBERSORTx, and Aginome-XMU. Neutrophils, fibroblasts, (naïve) B, and CD8+ T cells were sensitively identified by at least one method in both the coarse- (Fig. 6B) and fine-grained (Fig. 6C) sub-Challenges. All methods had low sensitivity in detecting CD4+ T cells (and their naïve and memory orientations) and macrophages, though several methods performed considerably better than others for both (CD4+ T cells: Aginome-XMU = 6%; D3Team = 7%; others ≥10% and macrophages: DA_505 = 8%; mitten_TDC19 = 8%; others ≥12%).

## Deconvolution methods quantify healthy and tumor-associated immune cells

The Challenge used immune cells isolated from healthy individuals, and methods were trained primarily on non-malignant expression profiles (Table 2). To assess whether these methods could also robustly detect immune cells derived from cancer samples, we generated in silico pseudo-bulk admixtures from published breast[34] and colorectal[35]

cancer scRNA-seq datasets (Methods). To do so, we mapped the cell subtypes identified in the breast and colorectal scRNA-seq studies to those predicted by Challenge deconvolution methods (Methods). For example, we mapped seven CD4 T cell subsets into the coarse-grained CD4 T cell population deconvolved by Challenge methods. Thus, our in silico admixtures reflect heterogeneity in real tumors.

Prediction performance was high for all cell types in both datasets, with at least one method achieving $r > 0.75$ on each population (Figs. S18, S19). Despite a shuffled ranking relative to that in the Challenge (Fig. S19; cf. Fig. 3A), top-performing methods performed well on average across the cell types in the two simulated cancer datasets (Fig. 7). Considering approaches that predicted all cell types in both cancer datasets, top-performing methods in the fine-grained sub-Challenge, mitten_TDC19 (Fig. S19; $r = 0.84$) and Aginome-XMU ($r = 0.76$) were outperformed in aggregate only by xCell ($r = 0.86$). The top-performing method in the coarse-grained sub-Challenge, CIBERSORTx, performed nearly as well relative to the cancer datasets ($r = 0.74$) as to the healthy Challenge dataset ($r = 0.79$).

There was considerable variability in cell type-level performance across healthy and cancer datasets (Fig. 7). To determine whether there was a systematic decrease in predicting cancer-associated immune cells relative to healthy immune cells, we quantified this variability across the two cancer datasets and the eight healthy immune cell datasets used in the Challenge via linear modeling (Methods). Method performance for specific cell types was generally not consistently lower in the two cancer datasets relative to healthy cell datasets (Fig. S20; Supplementary Data 16). Exceptions included NK cells, which were significantly lower in both the BRCA [$t(250) = -5.2$, Holm-Bonferroni(HB)-adjusted two-sided $p = 4.9 \times 10^{-5}$, effect size (model coefficient) = −0.13, 95% confidence interval (CI) = −0.17 to −0.08] and CRC [$t(250) = -8.9$, HB-adjusted two-sided $p = 2.0 \times 10^{-14}$, effect size = −0.22, 95% CI = −0.26 to −0.17] datasets, and neutrophils, which were significantly lower in the one cancer dataset including them [CRC; $t(125) = -8.1$, HB-adjusted two-sided $p = 5.6 \times 10^{-11}$, effect size = −0.19, 95% CI = −0.24 to −0.14]. In other

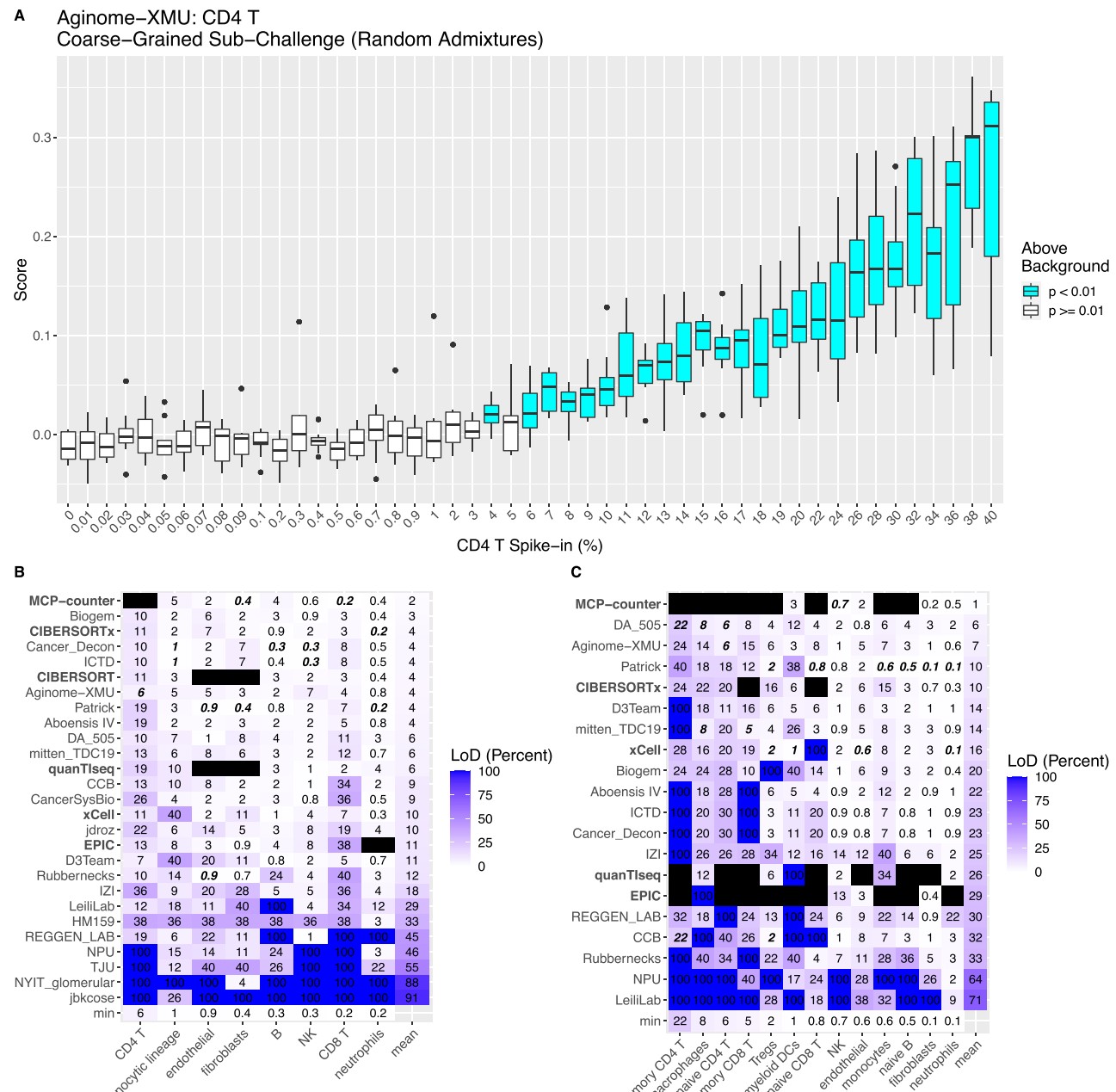

**Fig. 6 | Sensitivity of participant and comparator deconvolution methods.**
**A** Aginome-XMU predictions for CD4+ T cells (y axis) for unconstrained admixtures including the level of CD4+ T cells indicated (x axis). Limit of detection (LoD) is the least frequency at and above which all admixtures are above background (i.e., statistically different from the baseline admixture of 0% spike in based on a raw / uncorrected, two-sided Wilcoxon p value), which is 6% in this case. Boxplots display median (center line), 25th and 75th percentiles (hinges), and 1.5x interquartile range (whiskers) over n = 10 in silico unconstrained spike-in admixtures. Limits of detection for indicated methods (rows) and cell types (columns) calculated using n = 10 in silico unconstrained spike-in admixtures in each of the (**B**) coarse- and (**C**) fine-grained sub-Challenges. Best/lowest LoD for each cell type highlighted in bold italics. Methods ordered according to their mean LoD. Comparator methods in bold. Source data are provided as a Source Data file.

cases, differences in immune cell type performance were not consistent across both cancer-associated datasets, but rather significant in only one (Supplementary Data 16). Of note, heterogeneity across datasets also led to reduced performance for some cell types in some healthy datasets (Supplementary Data 16), and macrophage prediction performance was actually higher in both the BRCA [$t(150) = 9.4$, HB-adjusted two-sided $p = 1.4 \times 10^{-14}$, effect size = 0.35, 95% CI = 0.28–0.43] and CRC cancer datasets [$t(150) = 6.7$, HB-adjusted two-sided $p = 3.3 \times 10^{-10}$, effect size = 0.25, 95% CI = 0.18–0.33]. Taken together, our results suggest that deconvolution methods, evaluated in the Challenge using immune cells derived from healthy

patients, perform well in quantifying immune cell infiltration within tumors.

## Discussion

Computational deconvolution methods for unmixing gene expression profiles from pervasive bulk expression data can estimate cell type compositions, which have been shown to correlate with cancer phenotypes. The existing repositories of such bulk gene expression data describe large cohorts of patients through rich annotations, making them invaluable in addressing questions across biological domains. Additionally, bulk repositories continue to grow owing to difficulties in

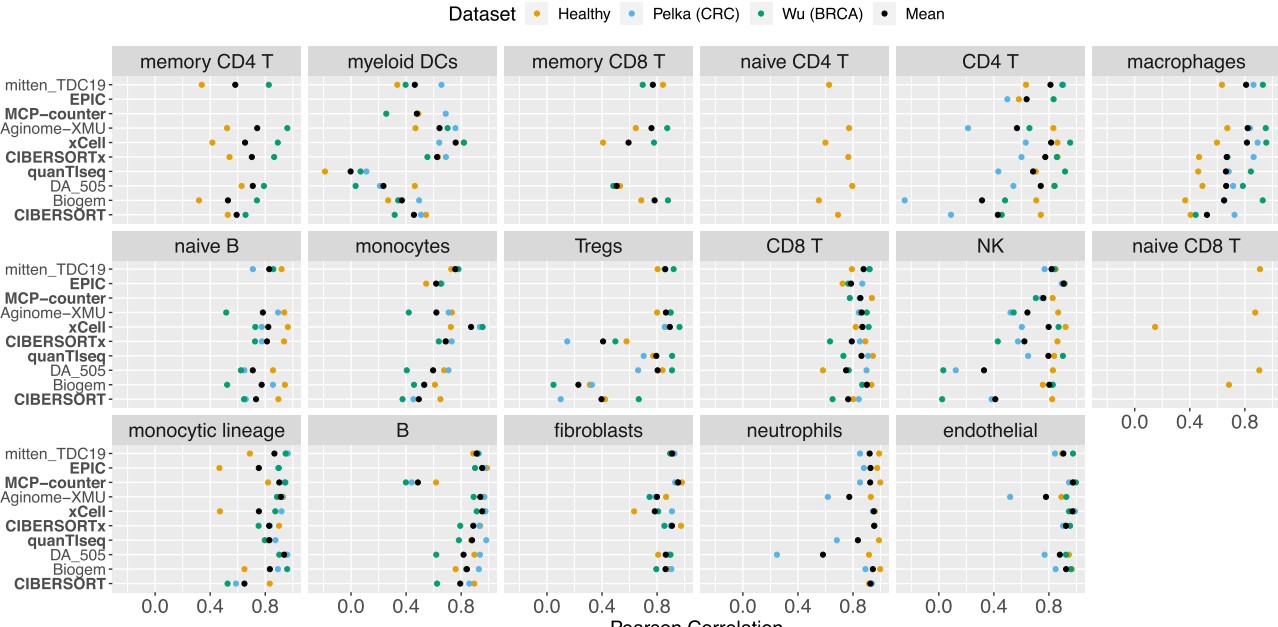

**Fig. 7 | Per-cell type performance of participant and comparator deconvolution methods across healthy and malignant datasets.** Performance (Pearson correlation; x axis) of methods (y axis) for each cell type (facet label). Methods include comparator baseline methods, participant methods ranking within the top three in either or both sub-Challenges, or methods having the best mean performance across datasets for any cell type. Performance indicated separately (by color) for Challenge validation (Healthy), in silico scRNA-seq-derived CRC [Pelka (CRC)], and in silico scRNA-seq-derived BRCA [Wu (BRCA)] datasets. Mean performance is calculated across these three datasets. Challenge validation performance is itself the mean performance across the eight healthy Challenge validation datasets (e.g., distinguished by in silico versus in vitro, as in Fig. S20). Methods ordered according to their mean performance across the three datasets and the cell types, and cell types ordered according to the max over methods of their mean performance across the three datasets. Source data are provided as a Source Data file.

applying single-cell techniques. Therefore, deconvolution of bulk expression data has and will continue to be an important tool in correlative studies, as noted by the large and increasing number of published methods. Unfortunately, benchmarking such methods has been hampered by the absence of ground truth data.

Here, we generated a resource and DREAM Challenge evaluation framework for a community assessment of deconvolution methods, through which we stimulated development of 22 new methods that we benchmarked alongside six published approaches. We did so using both in vitro and in silico admixtures of immune and stromal cells derived from healthy and, alternatively in the latter case, malignant tissues. We first isolated major immune and stromal populations likely to be found in tumors from healthy patients. We profiled these purified populations, as well as experimentally generated in vitro admixtures that included cancer cells, using RNA-seq. We further generated in silico admixtures of these purified non-malignant immune and stromal cells and of cancer-associated immune and stromal cells from published scRNA-seq studies. Using these, we assessed methods based on their ability to predict the comparative levels of an individual cell type across healthy (Fig. 3) and cancer (Fig. 7) samples (i.e., cross-sample, within-cell type) and further dissected the specificity (i.e., spillover; Fig. 5) and sensitivity (i.e., limit of detection; Fig. 6) of those per-cell type predictions. Combining these with prediction performance across cell types within an individual sample (i.e., cross-cell type, within-sample; Fig. 4) and with computational efficiency (Table 2) provides an overall summary of each method (Fig. S21). Two published methods, CIBERSORTx and MCP-counter, emerge with strong performance across these metrics for most of the cell types they evaluate within both healthy samples and tumor tissues. CIBERSORTx detects a large set of cell types and, as a reference-based method, computes proportions also comparable across cell types. MCP-counter identifies a more restricted set of cell types, but, owing to its enrichment-based approach, does so for each

independently. As such, it is applicable without complete knowledge of all cell types constituting a tissue. Notably, however, neither were trained to differentiate between memory and naïve CD8+ T cells, and xCell, the only published method that was, performs poorly in so doing. Here, participant methods Aginome-XMU, mitten_TDC19, Biogem, and DA_505 all show strong performance for memory and naïve CD8+ T cells (Fig. S21).

Aginome-XMU and xCell, another method performing well across test scenarios, enjoy the benefits typical of both reference- and enrichment-based approaches, allowing within-sample comparisons across cell types while being robust to contamination by unknown cell types not present in the training data, respectively. Aginome-XMU achieves this by predicting fractions using deep learning models, one for each cell type. Its success in our assessment demonstrates the relevance of deep learning to deconvolution – an alternative to reference- and enrichment-based approaches that does not require explicit identification of signatures or markers a priori. xCell achieves independence across cell types by instead calibrating its enrichment scores to a linear scale. The strong performance of the enrichment-based approaches mitten_TDC19, xCell, and MCP-counter, despite their computational simplicity, highlights the importance of marker selection (even if this occurs implicitly within Aginome-XMU). Regardless, no single method performed best on all cell types, nor did we observe that a particular high-level algorithmic approach dominated. This suggests that the best method may be problem specific and could be tailored to cell types of interest in a particular context. Our results provide the community cell type-level method performance to assist in making that decision. Moreover, we showed that a straightforward ensemble of all participant and comparator deconvolution methods marginally outperforms individual methods. This is consistent with findings from previous DREAM Challenges showing the superiority of such approaches[36–42]. As such, the ensemble may prove to be a robust strategy across cell types.

Deconvolution methods performed well in predicting levels of immune and stromal cells derived from tumor samples, despite their being trained largely on data derived from non-malignant samples. Our broad assessment over a wide range of immune populations buttresses those from a more focused evaluation, in which EPIC performance in predicting levels of several tumor-associated immune cells was similar across healthy- and tumor-derived signature matrices[14]. The authors did find differences in predictions of CD8+ T cells, which they attributed to a bias towards resting CD8+ T cells in the healthy samples. We did not observe this trend in our own assessment. We did note variability across validation datasets, but overall found no systematic evidence that methods trained on healthy samples underperformed in predicting tumor-associated immune cells. This robustness to training setting is consistent with practice. Certainly, transcriptomic[43] and proteomic[44] heterogeneity between healthy and tumor-associated immune cells has been reported, including the presence of tumor-specific activated macrophages. Nevertheless, the broad classification of immune cells (e.g., into CD8+ T cells) requires canonical markers that are shared across biological contexts - and their phenotypic isolation via FACS generally relies on them. Deconvolution methods leverage these shared markers, and indeed disease state-specific markers, including those of exhaustion, have been intentionally omitted in the training of some methods[14]. Analogously, different surface proteins are not used to sort CD4+ T cells as a function of what tissue they are in, though additional markers can further subclassify them.

Despite potential strides made by participant Challenge methods in predicting memory and naïve CD4+ T cells, macrophages, and monocytes, performance was lowest for these sub-populations. We can partially diagnose potential causes of this reduced performance by leveraging the sequenced purified populations to assess specificity, i.e., spillover from the purified population to others, and sensitivity, assessed as the limit of detection of in silico spike ins. We observed that memory CD4+ T cell predictions had poor specificity and sensitivity—ranking last in both metrics, with a median spillover across methods of 21%. Their best-case limit of detection was 22%, at which level they could be predicted above background by the most sensitive method. In contrast, other infiltrating immune cells had spillovers as low as ~5% for neutrophils and NK cells and could be detected down to a threshold of ~0.2% (neutrophils for CIBERSORTx and Patrick and CD8+ T cells for MCP-counter). Naïve and parental CD4+ T cell predictions also had poor sensitivity, with best-case limits of detection of 6%, and poor specificity, with spillovers of 15% and 16%, respectively. On one hand, poor prediction performance of macrophages may similarly be attributable to both low sensitivity, with a best-case limit of detection of 8%, and specificity indicated by 12% spillover. On the other hand, prediction performance of monocytes seems most likely due to low sensitivity (8% spillover) given the high specificity in detecting them, where the best-case limit of detection was ~1%. These difficulties in accurately predicting levels of CD4+ T cells and macrophages have important implications for tumor immunology given their importance as therapeutic targets[45].

Deconvolution methods, and our assessment of them here, are relevant despite the numerous experimental and computational approaches[2,3] for defining and characterizing the tissue micro-environment at single-cell resolution. These approaches present challenges slowing their broad adoption. Well-established techniques, such as FACS and immunohistochemistry, can rapidly and accurately count cells of specific types from a tissue, but are limited by the number of markers (and therefore cell types) they can simultaneously assay. Single-cell proteomic technologies, including mass cytometry (CyTOF), can quantify ~100 s of proteins in millions of cells at once, but require validated, high-quality antibodies for each marker of interest[46]. Single-cell RNA-seq (scRNA-seq) profiles the expression of individual cells[47], however artifacts are introduced by tissue preparation steps including dissociation and other manipulations, leading to preferential loss of specific cell types, such as plasma cells and neutrophils, as well as perturbation of cell state, with attendant transcriptome and proteome changes[8,48–51]. Such methods also cannot currently be applied to archival tissues and instead require prospective sample collection. In situ molecular imaging platforms, such as cyclic immunofluorescence[52], IMC[5], CODEX[4], and MIBI[6], can spatially resolve individual cells, but rely on predefined markers and appropriately prepared tissue. ST technologies are able to measure expression for thousands of genes, are applicable to formalin fixed paraffin embedded as well as fresh frozen tissue, and are rapidly advancing to single-cell resolution[7,53,54]. However, as with other emerging technologies above, they are costly and require specialized analyses that have yet to be standardized.

Our assessment has several limitations. (1) Our Challenge evaluates deconvolution methods according to the collective impact of their computational core (e.g., $v$-SVR or deep neural network), the manner in which they were trained, and the data upon which they were trained. As such, we can not attribute performance gains specifically to differences in algorithmic approach, reference signatures, or some combination. Further, many comparator methods could not be formally assessed by our two global metrics, since their published reference profiles did not include all the cell types required for scoring. Among such methods, CIBERSORTx was run on all cell types except naïve and memory CD8 T cells in the fine-grained sub-Challenge and exhibited comparable performance to other leading methods. MCP-counter could only predict half the specified cell types, but achieved high performance aggregated across those. Nevertheless, our results are relevant to the frequent practice[19–22,55–57] of applying a deconvolution method with its published signature and default settings. Further, we have assessed their cell type-specific performance. Finally, our purified expression profiles make possible the retraining of all methods on the same data to isolate the algorithmic component of their performance. In further support of such an evaluation, we have made code available to retrain all of the top-performing participant methods, Aginome-XMU (https://github.com/xmuyulab/DAISM-XMBD), Biogem (https://github.com/giannimonaco/DREAMChallenge_DeconvolutionTrain), DA_505 (https://github.com/martinguerrero89/Dream_Deconv_Challenge_Team_DA505_Training), and mitten_TDC19 (https://github.com/sdomanskyi/mitten_TDC19), while that for CIBERSORTx is already available via its online portal (https://cibersortx.stanford.edu). (2) The in vitro mixing of healthy immune cells and in silico mixing of scRNA-seq derived profiles of cancer-associated immune cells are individually imperfect, yet complementary. Isolating immune cells from peripheral blood, as for our in vitro admixtures, is more experimentally tractable than doing so from tumor tissues and avoids artifacts associated with scRNA-seq. Isolating cells from peripheral blood also allowed us to capture a broad array of cell types, including those that would otherwise be present at low frequency within tissues. Indeed, several populations were not represented in one or both of the two scRNA-seq studies used to generate our in silico admixtures, including neutrophils and memory and naïve CD4+ and CD8+ T cells. As such, not only were these particular cell types unevaluable in the cancer-associated admixtures, but a population that was present, such as memory CD4+ T cells, may have been easier to distinguish within them because closely related naïve CD4+ T cells were not included. Therefore, we believe that the healthy admixtures present a fuller, less biased representation for our evaluation. This is supported by overall consistent results with the cancer-associated in silico admixtures, despite the fact that the healthy cells do not reflect the full tumor-resident heterogeneity of immune populations. More generally, our approach of using in vitro derived or in silico simulated admixtures provides objective ground truth that would be difficult to otherwise obtain. In particular, the seemingly straightforward approach of using scRNA-seq-derived proportions as ground truth in evaluating deconvolution of bulk RNA-seq expression from matched samples is problematic: scRNA-seq experimental artifacts can skew cell type proportions

through preferential loss of certain cell types[8,48,58]. (3) Our Challenge does not capture all immune cell types of immuno-oncological significance, including those with exhaustion phenotypes[59]. Nevertheless, our fine-grained populations do extend the set of cell types previously evaluated. Further, that set includes cell types of clinical significance and that earlier deconvolution studies have already linked to survival[17,60,61], progression[62], and response to immune checkpoint inhibitors[19–21]. Finally, the top performers could be retrained to predict exhausted CD8+ T cells or other immune subpopulations characterized by single-cell studies by applying the code we make available here or the resources on the CIBERSORTx website. Indeed, CIBERSORTx was previously[29] trained to predict exhausted CD8+ T cells expressing *PDCD1* and/or *CTLA4* that were computationally isolated from a scRNA-seq study of melanoma[63]. Subsequent CIBERSORTx predictions of exhausted CD8+ T cells from bulk RNA-seq data were significantly correlated with response to immune checkpoint inhibitors across three independent melanoma studies[64–66]. (4) To maintain a practical scope, we have focused on deconvolution of mRNA expression. DNA methylation profiles have also proven to be a valuable source of information for tumor content deconvolution and it is possible to integrate multiple modalities to improve accuracy[67]. (5) Our focus in this study was on deconvolution of bulk RNA-seq, however ST technologies are rapidly emerging. To date, many of these approaches rely on sequencing small populations of cells across tissue regions–for example the multicellular regions "spots" assayed by the Visium platform[7] and the "pucks" of the Slide-seqV2 platform[68]. To accelerate development of ST algorithms, our purified expression profiles could be used to simulate data for assessment of ST deconvolution methods. ST deconvolution method development and assessment have considerations beyond that relevant in deconvolution of bulk RNA-seq data, including modeling of sparse (near single-cell) sequencing data, incorporating number of cells obtained from a matched histology image as a prior constraint, and sharing information across a spatial neighborhood. Others have reviewed[69] and benchmarked[70] methods in this field. We envision that the approaches outlined herein could be used in future studies as a blueprint to assess deconvolution algorithms tuned to ST data.

Our community-wide comparison of 28 community-contributed and published deconvolution methods revealed that levels of most major immune and stromal lineages were well predicted by most approaches. As such, our assessment suggests they provide robust signals for downstream correlative analyses. We observed considerable variability in predictive performance for minor lineages across methods. Though finer dissection was difficult for most subpopulations and most methods, even levels of the most challenging cell type, memory CD4+ T cells, were predicted at an accuracy ($r = 0.62$) that may be sufficient for some applications. Hence, our results allow researchers to choose the most appropriate method for studying an individual cell type. Where greater accuracy is needed, the purified immune and stromal expression profiles we generated should be a useful resource to the community in refining marker genes and signature matrices for deconvolution of the TME or of non-malignant contexts with significant immune modulation. This resource will also permit developers of new algorithms to perform unbiased comparisons, including for tools with increasing popularity that emerged after the Challenge competition was completed[71–73].

# Methods

## Ethical regulations
Fresh human whole blood was obtained from the Stanford Blood Center, in accordance with procedures approved by the Stanford Institutional Review Board (IRB, protocol #13942).

## Cell isolation
Cells were obtained from four sources, StemExpress (Folsom, CA), AllCells (Alameda, CA), ATCC (Manassas, VA), and the Human Immune

Monitoring Centering (HIMC) at Stanford University. Immune and stromal cells provided by StemExpress and AllCells were isolated according to vendor protocols (Supplementary Data 1). Cells provided by StemExpress had the following vendor IDs: CD8 T cells (PB08010C), Monocytes (PB14010C), Memory B cells (PB192701C), Naive B cells (PB1927N01C), Tregs (PB425002C), Naive CD4 T cells (PB445RA005C), Memory CD4 T cells (PB445RO05C), NK cells (PB56005C), Naive CD8 T cells (PB845RA005C), Eosinophils (PBEOS01C), and Macrophages (PBMAC001.5C).

Neutrophils and CD8+ memory T cells were isolated by HIMC as follows. Fresh human whole blood was obtained from the Stanford Blood Center, in accordance with procedures approved by the Stanford IRB. Human whole blood samples were collected under informed consent in EDTA-coated tubes. After 2 h resting, whole blood samples were split for neutrophils and CD8 memory T cell isolation.

Neutrophil isolation was performed with MACSxpress® Whole Blood Neutrophil Isolation Kit (Miltenyi Biotec; #130-104-434) according to manufacturer instructions. Briefly, the whole blood samples were mixed with the appropriate amount of isolation mix buffer consisting of magnetically coated beads conjugated to antibodies targeting all the immune populations in the peripheral blood except for the neutrophils. The cell suspension containing the isolation mix was incubated for 5 min at room temperature on a low-speed rotator. Then magnetic separation was performed for 15 min prior to collecting the untouched neutrophils in a clean tube.

For CD8+ memory T cell isolation, Peripheral Blood Mononuclear Cells (PBMCs) were first isolated by density gradient centrifugation using Ficoll-Paque™ Plus (Cytiva; #17144003). After washes, cell counts were obtained using a Vi-Cell XR cell viability analyzer (Beckman Coulter). Actual isolation was performed using a CD8+ Memory T Cell Isolation Kit (Miltenyi Biotec; #130-094-412) per the manufacturer's instructions. Briefly, PBMCs were incubated at 4 °C for 10 min with a cocktail of biotin-conjugated monoclonal antibodies against CD4, CD11c, CD14, CD15, CD16, CD19, CD34, CD36, CD45RA, CD56, CD57, CD61, CD123, CD141, TCRgd, and CD235a. After washing, cells were resuspended in a solution of anti-biotin magnetic microbeads and incubated for 15 min at 4 °C. After another wash, magnetic separation was performed using LS columns (Miltenyi Biotec; #130-042-401), and we collected the cell fraction corresponding to CD45RO+CD45RA-CD56-CD57-CD8+ T cells.

Finally, isolated neutrophils and CD8+ memory T cells were resuspended in RNAprotect Cell Reagent (Qiagen; #76526) for RNA extraction.

## Cell lines
Cell lines were obtained from ATCC for colon cancer cells (HT-29 colon adenocarcinoma; HTB-38), breast cancer cells (BT-483 breast ductal carcinoma; HTB-121), fibroblasts (normal dermal fibroblasts; PCS-201-012), and endothelial cells (primary aortic endothelial cells; PCS-100-11). We did not perform cell line authentication nor did we test for mycoplasma contamination. None of the cell lines are on the list of known misidentified cell lines maintained by the International Cell Line Authentication Committee (https://iclac.org/databases/cross-contaminations/).

## Library preparation, RNA sequencing, and data processing
Libraries were prepared using the Clontech SMARTer Stranded Total RNA-Seq v2 kit (Takara Bio) according to manufacturer instructions. Paired-end RNA sequencing of all in vitro admixtures and purified samples was performed by MedGenome Inc, by pooling the indexed libraries across four lanes of an Illumina NovaSeq S4 flowcell.

Estimated transcript read counts and transcripts per million (tpm) were generated via pseudo-alignment with Kallisto v0.46.0 to hg38 using Homo_sapiens.GRCh38.cdna.all.idx (ftp://ftp.ensembl.org/pub/release-92/fasta/homo_sapiens/cdna/Homo_sapiens.GRCh38.cdna.all.fa.gz). A translation table of Ensembl transcript ID to Ensembl gene ID

and gene symbol was derived using `biomaRt` and stored on the Synapse platform at syn21574276. Estimated gene read counts and tpm were calculated as the sum of transcript counts and tpm, respectively, associated with the gene via the translation table.

## Training data curation

Participants were provided with a curated list of purified expression profiles in GEO[74]. GEO annotations were queried using regular expressions corresponding to cell populations of interest (e.g., with patterns "T.reg", "regulatory", and "FOXP3" for regulatory T-cells). Specifically, GEO annotations for fields "source_name" or involving "characteristic" (e.g., "characteristics_ch1") were accessed via `GEOmetadb` in R[75]. Cell type patterns are available at: https://github.com/Sage-Bionetworks/Tumor-Deconvolution-Challenge/blob/master/scripts/training-data-curation/phenotypes-to-query.tsv. Matches identified via `grepl` were manually curated, resulting in tables associating cell populations with GEO samples. These were further summarized according to dataset, listing the cell populations and the number of cell populations represented by each dataset. This was intended to help participants prioritize datasets representing many or multiple cell types of interest. Per-sample and per-dataset tables were created separately for microarray (Supplementary Data 9, https://www.synapse.org/#!Synapse:syn18728081, and https://www.synapse.org/#!Synapse:syn18728088) and RNA-seq (Supplementary Data 10, https://www.synapse.org/#!Synapse:syn18751454, and https://www.synapse.org/#!Synapse:syn18751460) platforms. Microarray expression datasets were identified as those having "Expression profiling by array" in the "type" field of the "gse" SQLite table available in `GEOmetadb` and as being assayed in human [i.e., as having the pattern "sapiens" in the "organism" field of the "gpl" (platform) SQLite table]. RNA-seq expression datasets were similarly identified as having "Expression profiling by high throughput sequencing" in the "gse" table and as being assayed in human. Additionally, RNA-seq datasets were limited to those generated on Illumina platforms (i.e., as having a pattern of "illumina" in the "title" field of the "gpl" table), specifically the HiSeq (with pattern "hiseq" in the "title" field) or NextSeq (with pattern "nextseq" in the "title" field) platforms. Participants were provided only identifiers of GEO datasets (GSMxxx) and samples (GSExxx). In particular, cross dataset normalization to account for batch effects was not performed, but rather was left to participants.

## Unconstrained admixture generation

Unconstrained admixtures were defined in two stages: (1) a broken-stick approach partitioned the entire admixture across $n$ cell types and (2) the proportion of each cell type $c$ was restricted to be within $min_c$ and $max_c$. In particular, for $n$ cell types contributing a proportion $p <= 1$ (i.e., 100%) of the admixture total, the range 0 to $p$ was randomly broken into $n$ segments by choosing $n-1$ boundaries of those segments. The $n-1$ boundaries were uniformly sampled between a minimum cell population size of 0.01 (i.e., 1%) and $p$ in fixed-sized steps (of 0.01 unless otherwise specified), thus ensuring that each of the $n$ populations was represented at a frequency of at least 1%. The resulting candidate proportions were excluded if the proportion $p_c$ for any of the $n$ cell types $c$ was outside the bounds $[min_c, max_c]$. $min_c$ was set to 0 and $max_c$ was set to 1 (i.e., 100%), unless otherwise specified. Setting $p < 1$ allows the remaining $1-p$ proportion to be allocated to an $(n+1)^{st}$ cell type, e.g., fixing a spike in population at proportion $1-p$.

## Biological admixture generation

Biologically constrained admixtures were defined such that cell population proportions were within biologically reasonable limits, in particular those detected by CyTOF in PBMCs and aggregated in the 10,000 Immunomes (10KIP) database[76] (downloaded on July 9, 2018), observed in the Azizi single-cell (sc)RNA-seq breast cancer study[43], the

Tirosh scRNA-seq melanoma study[63], or inferred by CIBERSORT in the Thorsson TCGA pan-cancer study[77].

As none of these resources included all (coarse- or fine-grained) cell types to be deconvolved, several were combined in a hierarchical fashion. Two such hierarchical models, one based on the Thorsson study and the second based on the Azizi study, were created. At each level of the hierarchy, the models defined a minimum and maximum proportion for each population relative to its parental population. The minimum and maximum proportions for a particular cell type in a particular dataset were defined as two standard deviations above (~98th percentile) or below (~2nd percentile), respectively, the mean of proportions observed for that cell type in that dataset. The root of the model corresponds to the admixture of $n$ cell populations. In both hierarchical models, the entire admixture was partitioned into cancer cell, leukocyte, and non-leukocyte stromal compartments, with minimum and maximum proportions for each compartment defined using the Thorsson study. Specifically, from the stromal fraction (SF), or total non-tumor cellular fraction, and the leukocyte fraction (LF) defined by Thorsson, we define the cancer cell proportion as 1 - SF, the leukocyte proportion as LF, and the non-leukocyte stromal proportion as SF - LF. Both hierarchical models next subdivided the non-leukocyte stromal compartment into (cancer-associated) fibroblasts and endothelial cells using proportions of single cells observed in the Tirosh study. The original publication noted that four samples (CY58, CY67, CY72, and CY74) were experimentally enriched for immune infiltrates (CD45+). As such, the proportions inferred from them would not have represented cellular proportions relative to the entire cellular population. Hence, we excluded from analysis these samples, as well as CY75, which also did not have any tumor cells.

The Thorsson-based hierarchical model subdivided the leukocyte component into those inferred using CIBERSORT in the original study. Specifically, the leukocyte fraction was subdivided into the following sub-compartments: memory CD4+ T (i.e., T.Cells.CD4.Memory.Activated + T.Cells.CD4.Memory.Resting in the original publication), naïve CD4+ T (i.e., T.Cells.CD4.Naive), regulatory CD4+ T (i.e., T.Cells.Regulatory.Tregs), CD8+ T (i.e., T.Cells.CD8), memory B (i.e., B.Cells.Memory), naïve B (i.e., B.Cells.Naive), natural killer (i.e., NK_cells), neutrophils (i.e., Neutrophils), dendritic cells (i.e., Dendritic.Cells.Activated + Dendritic.Cells.Resting), monocytes (i.e., Monocytes), and macrophages (i.e., Macrophages.M0 + Macrophages.M1 + Macrophages.M2). Finally, the CD8+ T cell proportion was subdivided into memory and naïve CD8+ T cells using the 10KIP database.

The Azizi-based hierarchical model subdivided the leukocyte component into those reported in the Azizi study, specifically: T (i.e., T.cell in the original study), B (i.e., B.cell), natural killer (i.e., NK.cell), neutrophils (i.e., Neutrophil), dendritic cells (i.e., DC), monocytes (i.e., Monocyte), macrophages (i.e., Macrophage). Using the 10KIP database, the T cell compartment was further subdivided into memory, naïve, and regulatory CD4 T cells and memory and naïve CD8+ T cells, while the B cell compartment was further subdivided into memory and naïve B cells.

A single, final model was created from the Thorsson and Azizi models. This final model had a maximum proportion for cell type $c$ set to the maximum proportions for $c$ within the Thorsson and Azizi models. The final model's minimum proportion for $c$ was set to the minimum proportions for $c$ within the Thorsson and Azizi models, unless this was below 0.01, in which case it was set to 0.01.

The biologically constrained admixtures were generated using the Hit and Run Markov Chain Monte Carlo (MCMC) method for sampling uniformly from convex samples defined by linear (equality and inequality) constraints, as implemented in the `hitandrun` library in R. The system of linear constraints included a variable for each of the $n$ populations. As in the unconstrained admixtures, the corresponding $n$ proportions sum to $p <= 1$, with $p < 1$ allowing the remaining $1-p$ proportion to be allocated to an $(n+1)^{st}$ cell type. The

resulting equality constraint was passed to the `solution.basis` function, whose output was in turn passed to the `createTransform` function. *2n* linear inequality constraints were defined from the minimum and maximum proportions of each of the *n* populations. These were passed along with the output of `createTransform` to the `transformConstraints` function. An initial guess was created by passing these transformed constraints to `createSeedPoint` along with arguments `homogeneous=TRUE` and `randomize=TRUE`. Admixtures were sampled by passing the resulting seed and the transformed constraints to the `har` function along with parameters `N`, the number of iterations to run, set to $1000n^3$ and `N.thin`, the thinning factor indicating how many iterations to skip between samples, set to $n^3$.

### Selection of extremal candidate admixtures

Unless otherwise indicated, we ordered candidate admixtures so as to prioritize those most different from another. In particular, we select as the first two candidate admixtures those having maximum sum of squared (proportion) differences. Then, we greedily selected admixtures that maximized the minimal sum of squared differences to those admixtures already selected.

### In vitro validation admixture generation

60 biological admixtures and 36 unconstrained admixtures were defined using the procedures described in the subsections Biological admixture generation and Unconstrained admixture generation, respectively, with the exceptions noted below. Admixtures were defined over the cell populations having samples with sufficient mass and high RNA integrity number upon first assessment (Supplementary Data 2): breast or colorectal cancer, endothelial cells, neutrophils, dendritic cells, monocytes, macrophages, NK, regulatory T, naïve CD4+ T, memory CD4+ T, naïve CD8+ T, memory CD8+ T, naïve B, and memory B cells. Admixtures were designed so as to minimize batch effects across vendors, with half of the biological and half of the unconstrained admixtures assigned immune cells from Stem Express wherever availability allowed (Supplementary Data 3 and 4, respectively) and the rest assigned immune cells from AllCells wherever availability allowed (Supplementary Data 5 and 6). However, following subsequent experimental quantification, several cell populations (neutrophils, naïve CD8+ T cells, and memory B cells) did not have sufficient material for inclusion in the admixtures. As such, the final in vitro admixtures used during the Challenge validation phase included: breast or colorectal cancer cells, endothelial cells, fibroblasts, dendritic cells, monocytes, macrophages, NK, regulatory T, naïveCD4+ T, memory CD4+ T, memory CD8+ T, and naïve B cells. The final relative concentrations were rescaled relative to those designed computationally after excluding neutrophils, naïve CD8+ T cells, and memory B cells. The final in vitro admixtures used during the Challenge validation phase are provided in Supplementary Data 7 and 8.

Biological admixtures were generated with a fixed tumor proportion *1-p* in the range 0.2 to 0.8 in steps of 0.01 (i.e., such that the *n* populations excluding the tumor cells have proportions summing to *1-p*). This fixed tumor proportion overrode the tumor proportion bounds defined in the Thorsson-based and Azizi-based biological models.

To assess the ability of methods to differentiate between closely related signal/decoy pairs of cell types (e.g., memory vs naïve CD4+ T cells) and to improve our sensitivity in measuring this ability, within each unconstrained in vitro admixture we included a signal cell type with a high proportion ($min_c$ of 0.2 and $max_c$ of 0.35) and we excluded the decoy cell type ($min_c$ and $max_c$ of 0). For all other non-cancer cell types c, $min_c$ was set to 0.01 and $max_c$ to 0.5. We considered three ranges of cancer cell proportions: $min_{cancer} = 0.2$ to $max_{cancer} = 0.3$, $min_{cancer} = 0.4$ to $max_{cancer} = 0.5$ and $min_{cancer} = 0.6$ to $max_{cancer} = 0.7$. For each combination of these three cancer ranges and the following signal/decoy pairs, we generated 1000 candidate admixtures:

| Signal | Decoy |
|---|---|
| Monocytes | Dendritic cells |
| Macrophages | Monocytes |
| Dendritic cells | Macrophages |
| Naïve CD4+ T cells | Memory CD4+ T cells |
| Memory CD4+ T cells | Naïve CD4+ T cells |
| Naïve CD8+ T cells | Memory CD8+ T cells |
| Memory CD8+ T cells | Naïve CD8+ T cells |
| Naïve B cells | Memory B cells |
| Memory B cells | Naïve B cells |
| Tregs | Naïve CD4+ T cells |
| Naïve CD4+ T cells | Tregs |

Finally, we applied the strategy described in the subsection Selection of extremal candidate admixtures with a minor modification: in each selection round, we only considered candidate admixtures generated for a particular signal/decoy pair and we iterated through the list of pairs with each round (recycling pairs as necessary).

Code to generate the (unconstrained and biological) in vitro validation admixtures is in https://github.com/Sage-Bionetworks/Tumor-Deconvolution-Challenge/blob/master/analysis/admixtures/new-admixtures/gen-admixtures-061819.R. The admixture expression data (i.e., represented as TPM and as read counts) are in Synapse folder syn21821096 and the (ground truth) admixtures are in Synapse folder syn21820011. They are the datasets designated "DS1", "DS2", "DS3", and "DS4." Participants were told the cancer cell type included in each dataset (BRCA or CRC), which was BRCA for DS1 and DS3 and CRC for DS2 and DS4.

### In silico validation admixture generation

Insufficient RNA was available to include naïve CD8+ T cells and neutrophils in the in vitro admixtures. However, material was available to sequence the purified samples. This allowed us to generate in silico admixtures using the above biological and unconstrained procedures, such that the final in silico admixtures used during the Challenge validation phase included: breast or colorectal cancer cells, endothelial cells, fibroblasts, dendritic cells, monocytes, macrophages, neutrophils, NK, regulatory T, naïve CD4+ T, memory CD4+ T, memory CD8+ T, naïve CD8+ T, and naïve B cells. Memory B cells were unavailable to be included in either the in vitro or in silico admixtures.

For each of the two cancers (breast or colorectal) and each of the two vendor batches (i.e., Stem Express-enriched or AllCells-enriched, as described in the subsection In vitro validation admixture generation), we generated 15 fine-grained unconstrained admixtures and 20 fine-grained biological admixtures. Unconstrained admixtures were generated as described in the subsection Unconstrained admixture generation, except with a step size of 0.001. Further, we did not diversify admixtures by attempting to maximize the distance between them (as described in the subsection Selection of extremal candidate admixtures). We did diversify the biological admixtures, by generating each of the 20 admixtures in five batches and by applying the distance maximization procedure to select the four most distant admixtures from those in each batch of MCMC samples. We used these 140 fine-grained admixtures to generate 140 coarse-grained proportions by summing fine-grained proportions across sub-populations.

The transcripts per million (TPM)-based expression of in silico admixtures were generated as the weighted sum of the purified TPM expression profiles. For counts-based expression of the admixtures, we first normalized the gene counts for each purified sample by the total counts for that sample, multiplied by the median across samples

of sample total counts to obtain pseudo-counts on the same scale for each sample, and finally derived the admixtures as the weighted sum of the pseudo-counts.

Code to generate the (unconstrained and biological) in silico validation admixtures is in https://github.com/Sage-Bionetworks/Tumor-Deconvolution-Challenge/blob/master/analysis/validation_data/qc/generate-validation-in-silico-admixtures.R. The admixture expression data (i.e., represented as TPM and as read counts) and (ground truth) admixtures are in the same Synapse folders as the corresponding in vitro data—i.e., syn21821096 and syn21820011, respectively. They are the datasets designated "AA", "AB", "AE", and "AF." Participants were told the cancer cell type included in each dataset (BRCA or CRC), which was BRCA for AA and AE and CRC for AB and AF.

## Comparator method description

CIBERSORT[13] computes the linear combination of cell type expression profiles that optimally approximates the observed admixture expression over a set of markers, using ν-support vector regression (ν-SVR)[78,79], as implemented in `svm` of the R library `e1071`. CIBERSORT predicts 22 leukocyte populations, whose reference expression profiles across pre-defined markers are represented in the LM22 signature matrix. The optimization problem is solved in linear expression space, after the input admixture and the (vectorized) signature matrix are scaled to have zero mean and unit variance. All data for LM22 were obtained on the HGU133A microarray platform from healthy peripheral blood. Markers were defined following quantile normalization of the microarray expression data. Markers for a given cell type were defined so as to be: (1) differentially expressed between that cell type and all others; (2) amongst a subset of candidate markers, ordered by fold change, that optimized the condition number (i.e., quantified colinearity) of the resulting signature matrix; and (3) not expressed in non-hematopoietic healthy and malignant cells.

CIBERSORTx[29] uses the same, ν-SVR computational core as CIBERSORT. CIBERSORTx extends CIBERSORT, however, by correcting for differences between reference and input admixture data. In this Challenge, we applied bulk-mode (B-mode) batch correction, which re-optimizes the predicted fractions, β, relative to a ComBat-corrected[80] admixture a*. CIBERSORTx first solves for $\beta_0$ relative to input admixture **a** and signature matrix **S** (e.g., LM22), and then defines **a**\* by ComBat-correcting **a** so as to minimize its differences relative to the *estimated* admixture **S** $\beta_0$. The original CIBERSORTx publication defined several new signature matrices. As described below, we used one of these to first define the fraction of immune cells (relative to endothelial, stromal, and cancer cells) and then further subdivided this immune fraction across the 22 leukocytes of LM22.

EPIC[14] computes the linear combination of cell type expression profiles that optimally approximates the observed admixture expression over a set of markers, using constrained, weighted, least-squares optimization, as implemented in `constrOptim` of the R library `stats`. `constrOptim` minimizes a function subject to linear inequality constraints using an adaptive barrier algorithm. The inequality constraint on the sum allows for an uncharacterized cell type. EPIC operates in linear TPM expression space. Batch correction between reference and input admixture data consists of subsetting both datasets to a common set of genes and then normalizing the TPM expression values to sum to unity. EPIC collects cell type marker expression profiles in a signature matrix, which was derived using sorted immune cells from peripheral blood within three datasets of healthy patients following influenza vaccination and from healthy and non-malignant diseased patients. The authors performed no batch correction across the three studies, but noted that profiles segregated primarily by cell type, rather than study, using principal component analysis (Fig. 1 of the original study[14]). Markers were defined so as to be differentially expressed across reference cells and to be expressed by reference cells, but not within non-reference (or uncharacterized) cells (e.g., tumor). More specifically, differential

expression was performed using DESeq2[81] without specifying any batch correction and using pairwise comparisons of one reference cell type versus another. Markers were confirmed not to be expressed in non-hematopoietic tissues in the Illumina Human Body Map 2.0 Project (ArrayExpress ID: E-MTAB-513) or GTEx[82]. Finally, to avoid selecting for markers of exhaustion phenotype, for example, markers were required to have expression in reference samples and in tumor-infiltrating immune cells in non-lymphoid tissues[63].

MCP-counter[15] computes a cell type enrichment score as the arithmetic mean of the expression of that cell type's markers in linear expression space. The authors applied frozen robust multiarray average (fRMA) separately to each GEO series generated from three Affymetrix microarray platforms: Human Genome 133A, Human Genome 133 Plus 2.0, and HuGene 1.0 ST. The enrichment-based approach obviates the need for batch correction between reference and input admixture datasets. The authors organized cell populations hierarchically, and then for each population applied a stringent test requiring markers be differentially expressed between a population and populations not overlapping it in the hierarchy and further requiring that the marker not be expressed in the non-overlapping populations.

quanTIseq[17] computes the linear combination of cell type expression profiles that optimally approximates the observed admixture expression over a set of markers, using constrained least-squares optimization, as implemented in `lsei` of the R library `limSolve`. Unlike in the original publication, as applied in the Challenge (see below), quanTIseq does not scale inferred coefficients by mRNA content. quanTIseq operates in linear TPM expression space. quanTIseq collects cell type marker expression profiles in a signature matrix. Each immune marker was defined by a strategy requiring that it: (1) be expressed in at least two immune libraries; (2) be specific to its corresponding cell type (i.e., not a marker for a different cell type according to the xCell method); (3) have high (binned) expression in all libraries of that cell type and low or medium quantized expression in all other libraries; (4) not be highly expressed in non-hematological cell lines in the the Cancer Cell Line Encyclopedia (CCLE)[83]; (5) not be expressed in TCGA bulk tumors; (6) not be highly expressed; and (7) be highly correlated with mixing fractions simulated in 1700 datasets. The resulting signature matrix is composed of the median expression for the marker genes over all samples corresponding to a specific cell type.

xCell[18] computes a cell type score using ssGSEA[84] of the cell type's markers that it then calibrates to a linear scale. It deconvolves 64 cell types and uses spillover compensation to resolve those that are highly related. Markers were defined across 1725 non-malignant samples in six datasets generated using Cap Analysis Gene Expression (CAGE), RNA-seq, and microarrays. RNA-seq and CAGE data were FPKM normalized, while microarray data were normalized using robust multiarray average (RMA). Markers were defined independently in each dataset, which obviates the need for inter-technology batch correction. Markers for a specific cell type were selected such that a quantile of low expression within that cell type exceeded quantiles of high expression for all other cell types. Further, markers were required not to be expressed in CCLE carcinomas using a previously developed technique[85]. Enrichment scores were calculated using ssGSEA for each marker set within each dataset. The best three such signatures were assessed via validation in a held-out dataset and the mean of these was computed and fit to an analytical model that mapped it to cell type abundances.

## Comparator method evaluation

All comparator methods were executed by Sage Bionetworks (A.L., V.C., or B.S.W.) and without modification. All methods were passed expression in linear form.

CIBERSORT[13] was executed with arguments `abs_method = "sig.score"`, `absmean = TRUE`, `QN = FALSE`, and all other arguments default (including `absolute = FALSE`) via the script `CIBERSORT.R`.

Outputs from CIBERSORT were translated into Challenge populations as described in Table S9.

CIBERSORTx[29] was run in two phases: (1) The first phase separates immune cells (expressing *CD45*), endothelial cells (*CD31*), fibroblasts (*CD10*), and epithelial/tumor cells (*EPCAM*) using a signature matrix (Supplementary Table 2L of ref. [29]) derived from FACS purification of these four cell types within 26 surgically resected primary non-small cell lung cancer biopsies. (2) The second phase further divides the immune compartment into the 22 immune sub-populations represented by the same LM22 signature matrix originally published with CIBERSORT[13]. In both cases, CIBERSORTx was executed using the cibersortx/fractions docker container obtained from https://cibersortx.stanford.edu/, with arguments `--rmbatchBmode TRUE --perm 1 --verbose TRUE --QN FALSE`. The `--sigmatrix` parameter was used to specify the appropriate signature matrix. Outputs from the two phases of CIBERSORTx were translated into Challenge populations by scaling the output of LM22 phase by the CD45 output from the first phase as described in Table S10.

EPIC[14] was executed using the `EPIC` function from the `EPIC` R library, with the arguments `reference = "BRef"` and `mRNA_cell = FALSE`. Outputs from EPIC were translated into Challenge populations as described in Table S11.

MCP-counter[15] was executed using the `MCPcounter.estimate` function from the `MCPcounter` R library, with the argument `featuresType = 'HUGO_symbols'`. Outputs from MCP-counter were translated into Challenge populations as described in Table S12.

quanTIseq[17] was executed using the `deconvolute_quantiseq` function implemented in the `immundeconv` R library[24]. `deconvolute_quantiseq` was passed the arguments `tumor = TRUE`, `arrays = FALSE`, and `scale_mrna = FALSE`. If parameterization of `deconvolute_quantiseq` returned any invalid (i.e., not-a-number) results, it was re-run with the additional argument `method = "huber"`. Outputs from quanTIseq were translated into Challenge populations as described in Table S13.

xCell[18] was executed using the `xCellAnalysis` function of the `xCell` R library, with the argument `rnaseq = TRUE` and the argument `cell.types.use` set to the corresponding cell types within each challenge [i.e., to `c("B-cells", "CD4+ T-cells", "CD8+ T-cells", "NK cells", "Neutrophils", "Monocytes", "Fibroblasts", and "Endothelial cells")` in the coarse-grained sub-Challenge and to `c("Memory B-cells", "naive B-cells", "CD4+ memory T-cells", "CD4+ naive T-cells", "Treg", "CD8+ Tem", "CD8+ naive T-cells", "NK cells", "Neutrophils", "Monocytes", "DC", "Macrophages", "Fibroblasts", "Endothelial cells")` in the fine-grained sub-Challenge]. Outputs from xCell were translated into Challenge populations as described in Table S14.

## Deconvolution method scoring and comparison

Pearson and Spearman correlation-based scores were calculated hierarchically for a given method $a$: For each cell type $c$ and validation dataset $d$ (i.e., DS1, DS2, DS3, DS4, AA, AB, AE, and AF), the correlation between the values predicted by $a$ and the ground truth was calculated. These correlations were then averaged over all cell types $c$ to define the score of method $a$ for dataset $d$. These dataset-level scores were finally averaged over all datasets $d$ to define the aggregate score for method $a$.

To assess scoring differences in the primary metric between a top-performing method $a$ and another method $b$, we computed a Bayes factor $K_{a,b}$ over 1000 bootstrap samples and considered $K_{a,b} > 3$ as indicating a significant difference. More specifically, we bootstrap sampled (i.e., sampled with replacement) prediction scores separately within each dataset (i.e., DS1, DS2, DS3, DS4, AA, AB, AE, and AF), calculated a Pearson correlation-based score $S_i^a$ between the predictions in bootstrap sample $i$ for method $a$ and the corresponding ground truth values (and similarly for $S_i^b$ and method $b$), and calculated

$K_{a,b}$ as

$$K_{a,b} = \frac{\text{\# of bootstrap samples for which method } a \text{ outperformed method } b}{\text{\# of bootstrap samples for which method } b \text{ outperformed method } a}$$
$$= \frac{\sum_{i=1}^{1000} 1(S_i^a > S_i^b)}{\sum_{i=1}^{1000} 1(S_i^b > S_i^a)}$$

where $1(x)$ is the indicator function that equals 1 if and only if $x$ is true and is 0 otherwise. Any ties between methods $a$ and $b$ (i.e., $K_{a,b} \leq 3$) were resolved using the secondary Spearman correlation-based metric. However, this did not occur in the first submission results. Distributions, medians, and means over the $S_i^a$ are reported for the Pearson correlation-based scores in the figures (e.g., Fig. 2A) and main text in lieu of a single score on the original validation data. Similar bootstrapped distributions, medians, and means were calculated for the Spearman correlation-based scores and are likewise reported.

## Within-sample deconvolution method assessment

We assessed prediction performance across cell types within samples for those methods outputting normalized scores (CCB, D3Team, NYIT_glomerular), proportions (Patrick), and fractions (Aginome-XMU, Biogem, CIBERSORTx, DA_505, HM159, IZI, jbkcose, jdroz, LeiliLab, NPU, REGGEN_LAB, Rubbernecks, Tonys Boys, and xCell). We computed the Pearson correlation, Spearman correlation, and root-mean-square-error (RMSE) across cell types within a sample. To assess ties across teams, we fit a linear model whose response was the correlation or RMSE and whose dependent variable was the team. The top-scoring team (based on ordering of the median value across samples) was used as the reference in the linear model, which was fit using `lm` in R. Teams were considered tied with the top performer if their corresponding $t$-statistic $p$-value was >0.05, as computed from the model fit using `summary`.

Several outliers were excluded from the RMSE sub-plots of Fig. 4 (Patrick from Fig. 4A, B and NYIT_glomerular from Fig. 4A).

## Deconvolution method specificity assessment

To assess deconvolution method specificity, we calculated the (min-max) normalized prediction for a cell type $X$ in a sample $S$ purified for some cell type other $Y \neq X$. These normalized predictions are displayed in the heatmap of Fig. 5A, with cell types as columns and samples as rows. Predictions were normalized so as to be comparable across methods independent of the scale of the prediction (e.g., both unnormalized scores comparable across samples and proportions comparable across samples and cell types). The min-max normalization of a prediction *pred(X, S, m)* for cell type $X$, method $m$, and purified sample $S$ was defined as

$$[pred(X, S, m) - \min_{S'} pred(X, S', m)] / [\max_{S'} pred(X, S', m) - \min_{S'} pred(X, S', m)].$$

Spillover into (predicted) cell type $X$ for method $m$ was calculated as the above normalized prediction for cell type $X$ and method $m$ averaged over samples $S$ purified for some cell type $Y \neq X$ (i.e., the mean of the column corresponding to cell type $X$ in Fig. 5A that excludes elements in which $X$ is in the sample corresponding to the row). These spillovers were then averaged over sub-Challenges and the resulting distributions were plotted in Fig. 5B. Distributions of spillovers over cell types are plotted for each method in the coarse- (Fig. 5C) and fine-grained (Fig. 5D) sub-Challenges.

Code to format the purified expression profiles is in https://github.com/Sage-Bionetworks/Tumor-Deconvolution-Challenge/blob/master/analysis/specificity-analysis/create-spillover-dataset.R. The processed expression data (i.e., represented as TPM and as read counts) and the (ground truth) admixtures are in Synapse folder syn22392130.

## Deconvolution method sensitivity assessment

To assess deconvolution method sensitivity in detecting each cell type $X$, we generated in silico admixtures in which we computationally spiked in $X$ at regular proportions. We considered 49 spike-in levels from 0% to 0.1% in increments of 0.01%, from 0.1% to 1% in increments of 0.1%, from 1% to 20% in increments of 1%, and from 20% to 40% in increments of 2%. Cancer cells were neither used as spike ins nor included within the admixtures. Otherwise, all cell types with available purified expression profiles were included, namely endothelial cells, fibroblasts, dendritic cells, monocytes, macrophages, neutrophils, NK, regulatory T, naïve CD4+ T, memory CD4+ T, memory CD8+ T, naïve CD8+ T, and naïve B cells. Expression profiles of in silico admixtures were generated as described in the subsection In silico validation admixture generation.

We defined the limit of detection (LoD) for cell type $X$ and method $m$ as the least frequency at and above which $m$'s prediction for $X$ is statistically distinct from the baseline admixture (0% spike in). We assessed statistical significance using the two-sided Wilcoxon test as implemented in the `compare_means` function of the `ggpubr` library and using a raw (uncorrected) $p$-value cutoff of 0.01.

We generated both unconstrained and biological admixtures, using both fine- and coarse-grained populations. For unconstrained admixtures, we used the broken stick procedure described in the subsection Unconstrained admixture generation, except with a step size of 0.001 and without diversifying admixtures as described in the subsection Selection of extremal candidate admixtures. For each of the two vendor batches (i.e., Stem Express-enriched or AllCells-enriched, as described in the subsection In vitro validation admixture generation) and each spike in level $s$, we generated five coarse- and five fine-grained unconstrained admixtures such that the proportions of the $n$ populations summed to $1-s$. We used these same five admixtures for each of the spike-in experiments by simply assigning the population with fixed proportion $s$ the name of the population to be spiked in.

For unconstrained coarse-grained populations, we wanted to fix the level of the parental population (e.g., CD8+ T cells) rather than the sub-populations comprising it (i.e., memory and naïve CD8+ T cells). We defined coarse-grained admixtures at the level of the coarse-grained populations, but to concretely instantiate them we distributed the proportion of each parental population into its corresponding sub-populations. We did so by randomly dividing the proportion into $m$ sub-populations using a flat Dirichlet distribution (using the `rdirichlet` function in the `dirichlet` library) whose $m$ parameters were set to $1/m$.

Code to generate the in silico spike in admixtures is in https://github.com/Sage-Bionetworks/Tumor-Deconvolution-Challenge/blob/master/analysis/in-silico-admixtures/generate-in-silico-admixtures.R. The processed expression data (i.e., represented as TPM and as read counts) and the (ground truth) admixtures are in Synapse folder syn22361008.

## Statistical analyses

All analyses were performed using R statistical software[86]. Pearson and Spearman correlations were calculated with `cor.test` and two-sided Wilcoxon tests were performed using `compare_means`.

## Consensus rank method

We sought to define an ensemble method to aggregate predictions across all participant and comparator methods. Since the scales of predicted values vary according to the type of method output (scores, normalized scores, or fractions), we decided to aggregate the ranks of the predicted values across methods rather than the predicted values themselves. This is an instance of the consensus ranking, or social choice, problem in which we seek a ranking that summarizes the individual rankings of $n$ judges (or, in our case, methods) for $m$ objects (here, samples). We could define a consensus rank-based ensemble method using `ConsRank`[87], for example, which uses heuristic algorithms to define one or more consensus rankings. However, as the

(approximate) solutions are not guaranteed to be unique, we decided instead to take the more straightforward and more computationally efficient approach of simply defining the ensemble ranking as the mean of the individual rankings.

## Deconvolution method performance across healthy and cancer-associated immune cell types

We compared deconvolution performance across datasets independently for each cell type and after controlling for the mean performance of each method. Specifically, within each dataset, cell type, and method, we adjusted the Pearson correlation between the ground truth and predicted values by subtracting off the mean Pearson correlation across datasets for the respective cell type and method. We then performed an ANOVA by passing the formula `pearson.r ~ 0 + dataset` to `lm` in R. To test whether the Pearson correlation within any individual dataset was less than that of the mean across datasets, we computed one-sided $p$-values using the Student t distribution function `pt` with `lower=TRUE`. Finally, we adjusted all $p$-values calculated across all datasets and cell types for multiple hypothesis testing using the Holm-Bonferroni method implemented in `p.adjust`. We used the Holm-Bonferroni method as it makes no assumptions of the $p$-values, in particular, that they are independent, a condition that would not be met for our testing scenario. 95% confidence intervals were computed with `confint`.

## Pseudo-bulk Wu et al. BRCA dataset generation

We downloaded the raw count data and metadata generated by the Wu et al. BRCA single-cell study[34] from the Broad Single Cell Portal (https://singlecell.broadinstitute.org/single_cell). Specifically, we downloaded `Allcells_raw_count_out.zip` and `Whole_miniatlas_meta.csv` from https://singlecell.broadinstitute.org/single_cell/study/SCP1039/a-single-cell-and-spatially-resolved-atlas-of-human-breast-cancers. We extracted read counts from the raw data using `Read10X` from Seurat[88]. We mapped single cells as annotated in the metadata of the published study using the fields `celltype_major`, `celltype_minor`, and `celltype_subset` to those in the coarse-grained sub-Challenge as described in Table S15. The fine-grained cell types were subsequently mapped to coarse-grained cell types.

Separately for the coarse- and fine-grained translations, we defined the frequency of Challenge cell types within each patient. We further summed the raw gene counts for cells having the same annotation within a patient, resulting in raw counts for each cell type within a patient rather than for each cell. We excluded genes from the original study not present in the Challenge validation data. We excluded patients whose respective cells don't cluster with cells of the corresponding type from other patients. More specifically, we computed a reduced dimensionality UMAP projection from CPM-normalized deconvolution-related genes. If the most frequent annotation of a cell's ten nearest neighbors differed from its own annotation, that cell (type) was considered problematic and the corresponding patient was excluded from further processing. The following patients were excluded: CID4513 and CID4465 (because of mis-clustering of B cells), CID4461 (monocytic lineage), and CID4523 (endothelial cells). Deconvolution-related genes were those used in four published methods: MCP-Counter genes, CIBERSORT's LM22 genes, EPIC's TRef and BRef signature genes, and quanTIseq's TIL10 genes.

As some deconvolution methods expect Hugo gene symbols, whereas others expect Ensembl gene identifiers, we translated the former to the latter using the same translation table applied during the Challenge (available on Synapse via syn22394938). Where multiple Ensembl identifiers mapped to the same gene symbol, the associated raw counts were summed. Counts were normalized such that genes within each cell type/patient pair summed to the same value – namely, the maximum sum across all cell type/patient pairs. This facilitated generation of a pseudo-bulk sample for each patient by taking a sum of raw count, cell type expression profiles for the patient weighted by

 

their corresponding proportional frequency within that patient. These were then translated to CPM values and both the raw count and CPMs in both Ensembl and Hugo namespaces were supplied as input to Challenge participant and comparator methods. These were then executed against the data using the Dockerized methods submitted to the Challenge, with the exception of CIBERSORTx, which was run manually (using the same version and procedure as in the Challenge).

## Pseudo-bulk Pelka et al. CRC dataset generation

We generated pseudo-bulk CRC data from the Pelka et al. single-cell study[35], using the same procedure as described above for the Wu et al. study, except as noted below. We downloaded the raw data file `GSE178341_crc10x_full_c295v4_submit.h5` from GEO and the metadata file `crc10x_tSNE_cl_global.tsv` from the Broad Single Cell Portal (https://singlecell.broadinstitute.org/single_cell/study/SCP1162/human-colon-cancer-atlas-c295). We extracted the raw count data, with genes represented with Ensembl identifiers, using `Read10X_h5` with arguments `use.names=FALSE` and `unique.features=FALSE`. Using the `hdf5r` library, we further extracted the translation between Ensembl identifiers (`matrix/features/id` slot) and Hugo symbols (`matrix/features/name` slot). We removed genes from pseudoautosomal regions (i.e., with PAR_Y in their Ensembl identifier).

We mapped single cells as annotated in the metadata of the published study using the field `ClusterFull` to those in the fine-grained sub-Challenge as described in Supplementary Data 17. The fine-grained cell types were subsequently mapped to coarse-grained cell types.

## Statistics and reproducibility

1000 bootstraps were used to compare methods. A sample size of 1000 was chosen so that intra-method variance across bootstraps was small (i.e., relative to inter-method variance). No data were excluded. Cell populations were generally derived from two biological replicates (often from different vendors). Exceptions were memory B cells, which were not replicated owing to poor RNA integrity of the second sample. Further, fibroblasts, endothelial cells, and colon and breast cancer cells were not replicated, as they were derived from cell lines. The experiments were not randomized, since there were no experimental groups in our study and randomization was not relevant. The investigators were not blinded during the study, as there were no group allocations.

## Reporting summary

Further information on research design is available in the Nature Portfolio Reporting Summary linked to this article.

## Data availability

The raw RNA-seq data generated in this study have been deposited in the NCBI's Sequence Read Archive (SRA) database under accession code SRP365686. The processed RNA-seq data have been deposited in the NCBI's Gene Expression Omnibus[89] (GEO) database under accession code GSE199324. Additionally, the raw and processed RNA-seq data are hosted on the Synapse data-sharing platform at syn21557721 and syn21571479, respectively. Source data are provided with this paper and are available at https://doi.org/10.5281/zenodo.11246197. Raw count data and metadata generated by the Wu et al. BRCA single-cell study[34] can be downloaded from the Broad Single Cell Portal [https://singlecell.broadinstitute.org/single_cell/study/SCP1039/a-single-cell-and-spatially-resolved-atlas-of-human-breast-cancers]. Raw data generated by the Pelka et al. CRC single-cell study[35] are available under GEO accession code GSE178341.

## Code availability

Code used to generate all results, figures, and tables from this manuscript is available on GitHub (https://github.com/Sage-Bionetworks/Tumor-Deconvolution-Challenge/releases/tag/v1.0.0)[90] and at https://doi.org/10.5281/zenodo.11110924. The scripts used to generate each dataset, table,

and figure are indicated in Tables S16 and S17. Round one code implementations are available for the Aginome-XMU (https://github.com/xmuyulab/DCTD_Team_Aginome-XMU), Biogem (https://github.com/giannimonaco/DREAMChallenge_Deconvolution; https://github.com/giannimonaco/DREAMChallenge_DeconvolutionTrain), DA_505 (https://github.com/martinguerrero89/Dream_Deconv_Challenge_Team_DA505), and mitten_TDC19 (https://github.com/sdomanskyi/mitten_TDC19) participant methods. Training code are available for the Aginome-XMU (https://github.com/xmuyulab/DAISM-XMBD), Biogem (https://github.com/giannimonaco/DREAMChallenge_DeconvolutionTrain), DA_505 (https://github.com/martinguerrero89/Dream_Deconv_Challenge_Team_DA505_Training), and mitten_TDC19 (https://github.com/sdomanskyi/mitten_TDC19) methods. The script https://github.com/Sage-Bionetworks/Tumor-Deconvolution-Challenge/blob/master/analysis/validation-analysis/run-deconvolution-method-on-challenge-data.R demonstrates how to run a deconvolution method against the Challenge data, using xCell as an example, and to compare it to the Challenge results.

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

## Acknowledgements

B.S.W., A.L., X.G., T.Y., and J.G. were supported by an NCI Coordinating Center for the Cancer Systems Biology Consortium and the Physical Sciences-Oncology Network U24 grant (CA209923). A.J.G., A.M.N. (CA209971), and L.M.H (CA209988) were supported by NCI Cancer Systems Biology Consortium U54 grants, and J.S.R. by the German Federal Ministry of Education and Research (BMBF, Computational Life Sciences grant no. 031L0181B). The experimental data were generated under an administrative supplement from the CSBC to A.J.G. and J.G. S.D., T.B., and C.P. were supported by National Institutes of Health, grant no. R01GM122085. F.P and A.d.R were supported by the Programme Cartes d'Identité des Tumeurs (CIT) from the Ligue Nationale Contre le Cancer. A.J.G., H.M., and C.D. were supported by the Cancer Immune Monitoring and Analysis Consortium (CIMAC) through NIH grant 1U24CA224309. J.J.W. was supported by the SiRIC-Curie program (SiRIC grant INCa-DGOS- 4654).

## Author contributions

B.S.W., A.d.R., A.M.N., J.J.W., F.P., A.V., J.S.R., L.M.H., J.G., and A.J.G. designed the study and revised the manuscript. B.S.W. performed statistical analyses. A.L., V.C., Ji.B., T.Y., X.G., Ju.B. provided Challenge infrastructure. B.S.W., A.L., and V.C. executed deconvolution methods. S.S., and D.D. designed and produced figures. Y.L., R.Y., M.E.G.G., S.D., G.M., H.L., X.X., S.W., F.Z., W.Y., C.A.C., B.J.L., T.J.B., C.P., F.P.C., M.C., and members of the Tumor Deconvolution DREAM Challenge consortium developed community deconvolution methods and provided written summaries thereof. J.C., H.M., C.D., V.S., S.P., and J.E.L. provided and isolated CD8 T cells and neutrophils. B.S.W., L.M.H., and A.J.G. wrote the manuscript. All authors approved the final version of the manuscript.

## Competing interests

A.M.N. is a co-founder of CiberMed, Inc., and A.J.G. has consulted for CiberMed, Inc. J.S.R. received funding from GlaxoSmithKline and Sanofi and consultant fees from Travere Therapeutics and Astex Therapeutic. A.V. is currently employed by F. Hoffmann-La Roche Ltd. The remaining authors declare no competing interests.

## Additional information

[1]Sage Bionetworks, Seattle, WA, USA. [2]Centre de Recherche des Cordeliers, INSERM U1138, Université Paris Cité, Paris, France. [3]Institute for Stem Cell Biology and Regenerative Medicine, Stanford University, Stanford, CA, USA. [4]Department of Biomedical Data Science, Stanford University, Stanford, CA, USA. [5]INSERM U830 and Translational Research Department, Institut Curie, PSL Research University, Paris, France. [6]Programme Cartes d'Identité des Tumeurs, Ligue Nationale Contre le Cancer, Paris, France. [7]MRC Centre for Reproductive Health, the Queen's Medical Research Institute, University of Edinburgh, Edinburgh, UK. [8]Xiamen University, Xiamen, Fujian, China. [9]Institute of Biochemistry and Biotechnology, School of Medicine, National University of Cuyo, Mendoza, Argentina. [10]Michigan State University, East Lansing, MI, USA. [11]BIOGEM Institute of Molecular Biology and Genetics, Ariano Irpino, AV, Italy. [12]Department of Biomedical Engineering, Knight Cancer Institute, Oregon Health & Science University, Portland, OR, USA. [13]Heidelberg University, Faculty of Medicine, and Heidelberg University Hospital, Institute for Computational Biomedicine, Bioquant Heidelberg, Germany. [14]Department of Pathology, Cancer Hospital, Chinese Aacdemy of Medical Science, Beijing, China. [15]AmoyDx, Xiamen, Fujian, China. [16]Aginome Scientific, Xiamen, Fujian, China. [17]Laboratory of Intelligent Systems (LABSIN), Engineering School, National University of Cuyo, Mendoza, Argentina. [18]Department of Radiation Oncology, Beth Israel Deaconess Medical Center, Harvard Medical School, Boston, MA, USA. [19]Sylvester Comprehensive Cancer Center, Department of Public Health Sciences, University of Miami Miller School of Medicine, Miami, Florida, USA. [20]Stanford Functional Genomics Facility, Stanford University School of Medicine, Stanford, CA, USA. [21]Institute for Immunity, Transplantation, and Infection, Stanford University School of Medicine, Stanford, CA, USA. [22]Translational Applications Service Center, Stanford University School of Medicine, Stanford, CA, USA. [23]Department of Medicine, Stanford University School of Medicine, Stanford, CA, USA. [24]Department of Pathology, Stanford University, Stanford, CA, USA. [42]Present address: The Jackson Laboratory for Genomic Medicine, Farmington, CT, USA. * ✉e-mail: andrewg@stanford.edu

## Tumor Deconvolution DREAM Challenge consortium

Brian S. White [1,42], Aurélien de Reyniès[2], Aaron M. Newman [3,4], Joshua J. Waterfall [5], Andrew Lamb[1], Florent Petitprez [6,7], Yating Lin[8], Rongshan Yu [8], Martin E. Guerrero-Gimenez[9], Sergii Domanskyi[10], Gianni Monaco [11], Verena Chung [1], Jineta Banerjee[1], Daniel Derrick [12], Alberto Valdeolivas[13], Haojun Li[8], Xu Xiao [8], Shun Wang[14], Frank Zheng[15], Wenxian Yang [16], Carlos A. Catania[17], Benjamin J. Lang [18], Thomas J. Bertus[10], Carlo Piermarocchi[10], Francesca P. Caruso [11], Michele Ceccarelli [11,19], Thomas Yu[1], Xindi Guo[1], Julie Bletz[1], John Coller[20], Holden Maecker [21], Caroline Duault [21], Vida Shokoohi[20], Shailja Patel[22], Joanna E. Liliental[22], Stockard Simon[1], Aashi Jain[25], Shreya Mishra[25], Vibhor Kumar[25], Jiajie Peng[26], Lu Han[26], Gonzalo H. Otazu[27], Austin Meadows[28], Patrick J. Danaher[29], Maria K. Jaakkola[30,31], Laura L. Elo[30,32], Julien Racle[33,34], David Gfeller[33,34], Dani Livne[35], Sol Efroni[35], Tom Snir[35], Oliver M. Cast[36], Martin L. Miller[36], Dominique-Laurent Couturier[36], Wennan Chang[37], Sha Cao[38], Chi Zhang[39], Dominik J. Otto[40,41], Kristin Reiche[40,41], Christoph Kämpf[40], Michael Rade[40], Carolin Schimmelpfennig[40], Markus Kreuz[40], Alexander Scholz[40], Julio Saez-Rodriguez [13], Laura M. Heiser [12], Justin Guinney[1] & Andrew J. Gentles [4,23,24]✉

[25]Department of Computer Science, Indraprastha Institute of Information Technology, Okhla Ph-3, New Delhi, India. [26]School of Computer Science, Northwestern Polytechnical University, Xi'an, Shaanxi, China. [27]Center for Biomedical Innovation, New York Institute of Technology, College of Osteopathic Medicine, New York, USA. [28]Icahn School of Medicine at Mount Sinai, New York, USA. [29]NanoString Technologies, Seattle, WA, USA. [30]Turku Bioscience Centre, University of Turku and Åbo Akademi University, Turku, Finland. [31]Department of Mathematics and Statistics, University of Turku, Turku, Finland. [32]Institute of Biomedicine, University of Turku, Turku, Finland. [33]Department of Oncology UNIL CHUV, Ludwig Institute for Cancer Research, University of Lausanne, Lausanne, Switzerland. [34]Swiss Institute of Bioinformatics (SIB), Lausanne, Switzerland. [35]The Mina & Everard Goodman Faculty of Life Sciences, Bar-Ilan University, Ramat Gan, Israel. [36]Cancer Research UK, Cambridge Institute, University of Cambridge, Cambridge, UK. [37]Electrical and Computer Engineering, Purdue University, West Lafayette, IN, USA. [38]Department of Biostatistics, Indiana University, School of Medicine, Indianapolis, IN, USA. [39]Medical and Molecular Genetics, Indiana University School of Medicine, Indianapolis, IN, USA. [40]Department of Diagnostics, Fraunhofer Institute for Cell Therapy and Immunology, Perlickstraße 1, 04103 Leipzig, Germany. [41]Institute for Clinical Immunology, Leipzig University, Johannisallee 30, 04103 Leipzig, Germany.

