## [Peer Review File · Nature Communications]

Community assessment of methods to deconvolve cellular composition from bulk gene expressionEditorial Note: This manuscript has been previously reviewed at another journal that is not operating a transparent peer review scheme. This document only contains reviewer comments and rebuttal letters for versions considered at *Nature Communications*.

Reviewer #1 (Remarks to the Author):

The study delivered a bulk RNA-seq deconvolution benchmarking study among six widely used tools and 22 newly developed tools. Furthermore, the paper uses experimental and simulation data to generate ground truth for benchmarking tools in terms of coarser and finer cell types. I reviewed the revised paper, previous comments from other reviews, and corresponding responses from the authors. Overall, the paper delivered an interesting study for benchmarking bulk-RNA-seq deconvolution analysis tools. The author addresses most of the comments from previous reviewers. However, I still have some further concerns in response to reviewers 1 and 3. The below responses are my comments for the response letter.

Reviewer 1 Major comment 3: The reviewer criticized the deconvolution data should use healthy blood and tumors. The authors solved this comment by (1) introducing an example of deconvolution in tumors, (2) adding more simulation data, and (3) discussing the feasibility of training the model on health samples and using it for tumor cases. I agreed with the response logic, but the evidence of (2) and (3) cannot support the (1). (1) delivers two main points: deconvoluting bulk RNA-seq can be used for immune checkpoint inhibitor effectiveness and patient prognosis prediction. Therefore, (2) should use scRNA-seq data labeled with immune checkpoint inhibitor and tumor prognosis to simulate bulk data. Then, compare which software produces results that align with clinical outcome predictions. Additional discussions in (3) should mention how to improve the precision and generalizability of simulated data based on the current large number of pan-cancer studies at the single-cell level, especially considering the abundance of cell type in scRNA-seq across various cancers and aiming to enhance the clinic usage for deconvolution for pan-cancer.

Reviewer 1 major comment 4: The reviewer criticized the insufficient cell types in immune-oncology for benchmarking. The author discusses the limitations of lacking such cell types. Actually, in the current immune-oncology research, exhausted CD8 T cells (progenitor, intermediate, and terminal state) are extremely important as they are directly associated with immunotherapeutic efficacy and cell therapy. Merely discussion in the paper is not sufficient to convince immuno-oncologists to use deconvolution in analyzing cell proportion in tumor microenvironment. Therefore, the author should at least simulate data with a certain number of exhausted CD8 T cells to benchmark deconvolution tools.

Reviewer 3 major comment 3: the reviewer was asking for using matched bulk-RNA-seq and matched scRNA-seq (i.e., bulk-RNA-seq and scRNA-seq data are from the same sample) to benchmark tools in real situations. The author used scRNA-seq to simulate data for solving this comment. However, the response does not fully address the reviewer's question. From the description, the BRCA and CRC cancer scRNA-seq are used to simulate pseudo-bulk RNA-seq. However, Fig. 7 did not include all cell types (e.g., four CD4 T cell subsets, five CD8 T cell subsets, and other immune cells in breast cancer) described in the original paper. Therefore, the results cannot support the real cell portion in tumors.

My minor comment in addition to previous reviewers:

The color in Fig.3A does not fit common coloring strategies. Usually, red color means high correlation, and blue means low correlation. If the author can make a change, it will be better for potential readers.

Reviewer #2 (Remarks to the Author):

With regard to responses to Reviewer #2's concerns:

1. Description of the new Methods:

While the mathematical details provided in Table 1 and Table 2 are valuable, they may not be immediately intuitive for the broader biology community. Would it be feasible to succinctly summarize each method in a single sentence just like how marker selection is delineated in Table 1 and Aginome-XMU's objective function?

2. Perspectives on deconvolution: As the field of genomics evolves, there emerges a key application of deconvolution which is about spatial transcriptomics. 10X Visium platform is a major one for measuring bulk transcriptomic profiles of cell aggregates on tissue slides and relies on deconvolution for interpretation. However, more fundamental deconvolution techniques like Seurat

still dominate the field, primarily due to their user-friendliness. Benchmarking deconvolution methods could potentially spearhead progress in spatial transcriptomic analyses, fostering a deeper understanding of tissue microenvironments and pathological processes. I recommend that the authors acknowledge this trajectory and provide insights into how their study could contribute to this burgeoning area.

Additional comments:

1. This is a useful benchmarking study on a key topic of genomics - reference-based deconvolution of bulk transcriptomic data. However, it's a missed opportunity that established methods like MuSiC and prominent techniques in spatial transcriptomics (e.g., Seurat and cell2location) weren't factored into the comparative analysis. Including these could substantially enhance the impact of the paper.

2. Figure 1a's legend shouldn't use "boxes shaded blue" to refer to the coarse-grained cell types because the 8 blocks are not all rectangle boxes. Maybe "blocks" or "segments" rather than "boxes"?

3. The diagram in Figure 1b (pertaining to the in vitro section) doesn't resonate intuitively. The text said that it's the RNAs that were mixed physically, but the diagram showed that cells were mixed and placed in culture dishes before sequencing, which seems incongruent with the actual experimental protocol and warrants reconsideration.

Reviewer #3 (Remarks to the Author):

Dear editor and authors,

In my opinion, the authors did address most of the raised points of concern very well. A benchmarking focused on both coarse and fine grained annotation is an added value to the public. The generated data is important for future tool development to allow for proper benchmarking against what is existing.

Since I was not part of the initial submission, I still have some remarks.

Which of the methods hold proportional fold changes between samples? In some cases that might be more important than to predict the exact percentages themselves.

The paper might benefit from an overview figure, summarizing the metrics from the individual analysis and figure panels, so it is easier to compare the methods.

The created data will become available, which is good for future deconvolution tool releases to benchmark. It would be even better if the authors provide a script/tool to compare their own results with these in the DREAM challenge.

I do have some additional minor remarks.

In the overview Figure 1 and from the result section (and even the methods) it is not fully clear to me to which level immune cells were in vitro purified?

In Figure 2: I would make the x-axis comparable for both coarse and fine grained annotation to ease interpretation.

In multiple figures cell types are sorted based on alphabet. It would be handier for interpretation if T cell subtypes are all together instead of spread over the graph.

It is not fully clear to me why CD4/CD8T cells are still a separate category while the subtypes are also included.

Kind regards

Reviewer #4 (Remarks to the Author):

I co-reviewed this manuscript with one of the reviewers who provided the listed reports as part of the Nature Communications initiative to facilitate training in peer review and appropriate recognition for co-reviewers.

Reviewer #5 (Remarks to the Author):

I co-reviewed this manuscript with one of the reviewers who provided the listed reports as part of the Nature Communications initiative to facilitate training in peer review and appropriate recognition for co-reviewers.

Point-by-point response

We appreciate the constructive comments from the Nature Communications reviewers. We are pleased that they feel we have addressed most of the major concerns raised by the initial round of reviews from Nature Methods. We believe our significant revisions in response to their latest comments improved the readability and presentation of our manuscript.

Our major revisions include:

1. Discussing published work that relates deconvolution predictions to downstream clinical phenotypes, including how predicted levels of exhausted CD8 T cells correlate with response to checkpoint inhibitors.
2. Discussing how the code we provide could be used to retrain our top-performing methods to predict cell types beyond those assessed in this Challenge, including exhausted CD8 T cells and other subtypes of T cells.
3. Adding a brief description of each comparator deconvolution method.
4. Adding a perspective on deconvolution of spatial transcriptomics data.
5. Completely redesigning our overview **Fig 1** to more accurately describe the Challenge setup and to more clearly indicate the many comparisons performed in the manuscript.
6. Providing a script that demonstrates how to run a deconvolution method against our Challenge data and compare its performance with results from the Challenge.

Below, we include our point-by-point response to each reviewer comment. Their original comments are highlighted with a black vertical bar in the margin. Our responses follow each, with any revised text indented in blue.

Reviewer 1

The study delivered a bulk RNA-seq deconvolution benchmarking study among six widely used tools and 22 newly developed tools. Furthermore, the paper uses experimental and simulation data to generate ground truth for benchmarking tools in terms of coarser and finer cell types. I reviewed the revised paper, previous comments from other reviews, and corresponding responses from the authors. Overall, the paper delivered an interesting study for benchmarking bulk-RNA-seq deconvolution analysis tools. The author addresses most of the comments from previous reviewers. However, I still have some further concerns in response to reviewers 1 and 3. The below responses are my comments for the response letter.

We appreciate Reviewer 1's diligence in carefully reading our manuscript and the earlier review correspondences. We also thank them for noting that we have addressed the majority of the comments from previous reviewers.

Reviewer 1 Major comment 3: The reviewer criticized the deconvolution data should use healthy blood and tumors. The authors solved this comment by (1) introducing an example of deconvolution in tumors, (2) adding more simulation data, and (3) discussing the feasibility of

training the model on health samples and using it for tumor cases. I agreed with the response logic, but the evidence of (2) and (3) cannot support the (1). (1) delivers two main points: deconvoluting bulk RNA-seq can be used for immune checkpoint inhibitor effectiveness and patient prognosis prediction. Therefore, (2) should use scRNA-seq data labeled with immune checkpoint inhibitor and tumor prognosis to simulate bulk data. Then, compare which software produces results that align with clinical outcome predictions. Additional discussions in (3) should mention how to improve the precision and generalizability of simulated data based on the current large number of pan-cancer studies at the single-cell level, especially considering the abundance of cell type in scRNA-seq across various cancers and aiming to enhance the clinic usage for deconvolution for pan-cancer.

We appreciate that the Reviewer agrees with the logic of our revisions to the manuscript. They also raise the additional point that robust tumor deconvolution may enable clinical decision making and prognosis prediction. We have modified the Discussion to address the additional comments from this reviewer. Specifically, we discuss the benefits of simulation to address Reviewer point (1):

More generally, our approach of using *in vitro* derived or *in silico* simulated admixtures provides objective ground truth that would be difficult to otherwise obtain. In particular, the seemingly straightforward approach of using scRNA-seq-derived proportions as ground truth in evaluating deconvolution of bulk RNA-seq expression from matched samples is problematic: scRNA-seq experimental artifacts can skew cell type proportions through preferential loss of certain cell types.^{8,48,58}

As rationale that deconvolution methods can indeed deconvolve real tumors [Reviewer point (1)] and that these results are correlated with clinical outcome [Reviewer point (2)], including response to checkpoint inhibition [Reviewer point (3)], we cite several published cases:

Our Challenge does not capture all immune cell types of immuno-oncological significance, including those with exhaustion phenotypes.⁵⁹ Nevertheless, our fine-grained populations do extend the set of cell types previously evaluated. Further, that set includes cell types of clinical significance and that earlier deconvolution studies have already linked to survival,^{17,60,61} progression,⁶² and response to immune checkpoint inhibitors.¹⁹⁻²¹

We conclude by addressing the use of pan-cancer scRNA-seq datasets to improve generalizability of deconvolution and how this has been done in correlating deconvolution predictions to checkpoint inhibition response:

Finally, the top performers could be retrained to predict exhausted CD8+ T cells or other immune subpopulations characterized by single-cell studies by applying the code we make available here or the resources on the CIBERSORTx website. Indeed, CIBERSORTx was previously²⁹ trained to predict exhausted CD8+ T cells expressing *PDCD1* and / or *CTLA4* that were computationally isolated from a scRNA-seq study of

melanoma.⁶³ Subsequent CIBERSORTx predictions of exhausted CD8+ T cells from bulk RNA-seq data were significantly correlated with response to immune checkpoint inhibitors across three independent melanoma studies.^{64–66}

Together, these numerous citations provide evidence that deconvolution results are relevant to clinical outcome. Performing our own correlative studies is beyond the scope of the current Challenge and manuscript, however we envision that it may serve as an interesting community-wide challenge to explore in future studies.

For the Reviewer's convenience, we here include the above references:

8. Slyper, M. *et al.* A single-cell and single-nucleus RNA-Seq toolbox for fresh and frozen human tumors. *Nat. Med.* **26**, 792–802 (2020).
17. Finotello, F. *et al.* Molecular and pharmacological modulators of the tumor immune contexture revealed by deconvolution of RNA-seq data. *Genome Med.* **11**, 34 (2019).
19. Petitprez, F. *et al.* B cells are associated with survival and immunotherapy response in sarcoma. *Nature* **577**, 556–560 (2020).
20. Helmink, B. A. *et al.* B cells and tertiary lymphoid structures promote immunotherapy response. *Nature* **577**, 549–555 (2020).
21. Nabet, B. Y. *et al.* Noninvasive Early Identification of Therapeutic Benefit from Immune Checkpoint Inhibition. *Cell* **183**, 363–376.e13 (2020).
29. Newman, A. M. *et al.* Determining cell type abundance and expression from bulk tissues with digital cytometry. *Nat. Biotechnol.* **37**, 773–782 (2019).
48. Denisenko, E. *et al.* Systematic assessment of tissue dissociation and storage biases in single-cell and single-nucleus RNA-seq workflows. *Genome Biol.* **21**, 130 (2020).
58. Maden, S. K. *et al.* Challenges and opportunities to computationally deconvolve heterogeneous tissue with varying cell sizes using single-cell RNA-sequencing datasets. *Genome Biol.* **24**, 288 (2023).
59. Pauken, K. E. & Wherry, E. J. Overcoming T cell exhaustion in infection and cancer. *Trends Immunol.* **36**, 265–276 (2015).
60. Cindy Yang, S. Y. *et al.* Pan-cancer analysis of longitudinal metastatic tumors reveals genomic alterations and immune landscape dynamics associated with pembrolizumab sensitivity. *Nat. Commun.* **12**, 5137 (2021).
61. Mandal, R. *et al.* The head and neck cancer immune landscape and its immunotherapeutic implications. *JCI Insight* **1**, e89829 (2016).
62. Xue, W. & Shi, J. Identification of genes and cellular response factors related to immunotherapy response in mismatch repair-proficient colorectal cancer: a bioinformatics analysis. *J. Gastrointest. Oncol.* **13**, 3038–3055 (2022).
63. Tirosh, I. *et al.* Dissecting the multicellular ecosystem of metastatic melanoma by single-cell RNA-seq. *Science* **352**, 189–196 (2016).
64. Chen, P.-L. *et al.* Analysis of Immune Signatures in Longitudinal Tumor Samples Yields Insight into Biomarkers of Response and Mechanisms of Resistance to Immune Checkpoint Blockade. *Cancer Discov.* **6**, 827–837 (2016).
65. Van Allen, E. M. *et al.* Genomic correlates of response to CTLA-4 blockade in metastatic melanoma. *Science* **350**, 207–211 (2015).
66. Nathanson, T. *et al.* Somatic Mutations and Neoepitope Homology in Melanomas Treated with CTLA-4 Blockade. *Cancer Immunol Res* **5**, 84–91 (2017).

Reviewer 1 major comment 4: The reviewer criticized the insufficient cell types in immune-oncology for benchmarking. The author discusses the limitations of lacking such cell types. Actually, in the current immune-oncology research, exhausted CD8 T cells (progenitor, intermediate, and terminal state) are extremely important as they are directly associated with immunotherapeutic efficacy and cell therapy. Merely discussion in the paper is not sufficient to convince immuno-oncologists to use deconvolution in analyzing cell proportion in tumor microenvironment. Therefore, the author should at least simulate data with a certain number of exhausted CD8 T cells to benchmark deconvolution tools.

We acknowledge that tumor ecosystems are comprised of additional cell types and states beyond those we included in this study. However, comprehensive assessment of all cell types and states was not feasible, so we prioritized a subset of clinically-relevant cell types with consensus surface markers that could be used for their purification. Importantly, many groups have successfully used deconvolution approaches to disentangle the cell types we prioritized – indeed, this was a key motivation for this DREAM Challenge. We now cite numerous published studies in which deconvolution of the cell types used in our Challenge have been shown to relate to clinical endpoints, including response to checkpoint inhibitors:

Further, [the set of cell types assessed in the Challenge] includes cell types of clinical significance and that earlier deconvolution studies have already linked to survival,^{17,60,61} progression,⁶² and response to immune checkpoint inhibitors.^{19–21}

We agree with the reviewer that CD8 T cells are an important and diverse population of cells in tumor ecosystems. A complete evaluation of exhausted CD8 T cells would require—in addition to simulation—our worldwide participants to retrain their methods on new data, and re-engaging them is not feasible at this time. However, prior studies provide strong evidence for the concept that deconvolution methods benchmarked in this study have the power to deconvolute exhausted CD8 T cells. Specifically, CIBERSORTx can be used to infer levels of exhausted CD8 T cells that were then found to correlate with response to checkpoint inhibitors. We have modified the discussion to include this important point:

CIBERSORTx was previously²⁹ trained to predict exhausted CD8+ T cells expressing *PDCD1* and / or *CTLA4* that were computationally isolated from a scRNA-seq study of melanoma.⁶³ Subsequent CIBERSORTx predictions of exhausted CD8+ T cells from bulk RNA-seq data were significantly correlated with response to immune checkpoint inhibitors across three independent melanoma studies.^{64–66}

Reviewer 3 major comment 3: the reviewer was asking for using matched bulk-RNA-seq and matched scRNA-seq (i.e., bulk-RNA-seq and scRNA-seq data are from the same sample) to benchmark tools in real situations. The author used scRNA-seq to simulate data for solving this comment. However, the response does not fully address the reviewer's question. From the description, the BRCA and CRC cancer scRNA-seq are used to simulate pseudo-bulk RNA-seq. However, Fig. 7 did not include all cell types (e.g., four CD4 T cell subsets, five CD8 T cell

subsets, and other immune cells in breast cancer) described in the original paper. Therefore, the results cannot support the real cell portion in tumors.

We acknowledge that the *predictions* reported in **Fig. 7** do not include all cell types (particularly, T cell subpopulations) described in the original paper. However, the simulated bulk admixtures that were deconvolved *do* contain a heterogeneity of subpopulations beyond those assessed in the Challenge. We have clarified these important points in the manuscript:

To do so, we mapped the cell subtypes identified in the breast and colorectal scRNA-seq studies to those predicted by Challenge deconvolution methods (see Methods). For example, we mapped seven CD4 T cell subsets into the coarse-grained “CD4 T cell” population deconvolved by Challenge methods. Thus, our *in silico* admixtures reflect heterogeneity in real tumors.

While our study is imperfect in assessing the *growing* universe of cell types, we do believe it includes cell types of clear (and referenced) clinical significance.

My minor comment in addition to previous reviewers:

The color in Fig.3A does not fit common coloring strategies. Usually, red color means high correlation, and blue means low correlation. If the author can make a change, it will be better for potential readers.

Thank you. We have switched red to indicate high correlation and blue to indicate low, as the Reviewer requested, in **Figs 3A, S7, S9, S11, S14, S18, and S19**.

Reviewer 2

1. Description of the new Methods:

While the mathematical details provided in Table 1 and Table 2 are valuable, they may not be immediately intuitive for the broader biology community. Would it be feasible to succinctly summarize each method in a single sentence just like how marker selection is delineated in Table 1 and Aginome-XMU's objective function?

We agree with the Reviewer that including a succinct description of methods will enhance the utility of our manuscript as a resource for further methods development. We added brief descriptions of the comparator methods at the end of the first Results section:

... we applied six widely used published tools (CIBERSORT,¹³ CIBERSORTx,²⁹ EPIC,¹⁴ MCP-counter,¹⁵ quanTIseq,¹⁷ and xCell¹⁸) as baseline comparator methods (see **Tables 1-2** and **Methods section Comparator method description**). Briefly, CIBERSORT¹³ computes the linear combination of cell type expression profiles that optimally approximates the observed admixture expression over a set of markers, using ν -support vector regression (ν -SVR). CIBERSORTx²⁹ uses the same, ν -SVR computational core as CIBERSORT, but additionally corrects for differences between reference and input

admixture data. EPIC¹⁴ computes the linear combination of cell type expression profiles that optimally approximates the observed admixture expression over a set of markers, using constrained, weighted, least-squares optimization. MCP-counter¹⁵ computes a cell type enrichment score as the arithmetic mean of the expression of that cell type's markers in linear expression space. quanTIseq¹⁷ computes the linear combination of cell type expression profiles that optimally approximates the observed admixture expression over a set of markers, using constrained least-squares optimization. xCell¹⁸ computes a cell type score using single-sample gene set enrichment analysis of the cell type's markers that it then calibrates to a linear scale.

This complements the description of the top-performing methods included later in the Results section:

Aginome-XMU, published subsequent to the Challenge³⁰, utilizes a neural network composed of an input layer, five fully connected hidden layers, and an output layer (Supplemental Methods; https://github.com/xmuyulab/DCTD_Team_Aginome-XMU; **Table S13**). The network effectively applies feature selection automatically and was trained here using synthetic admixtures. DA_505 applies a rank-based normalization, selects features by applying random forests to synthetic admixtures, and ultimately applies regression to predict abundance of each cell type *independently* (Supplemental Methods; https://github.com/martinguerrero89/Dream_Deconv_Challenge_Team_DA505; **Fig. S2**; **Table S14**). mitten_TDC19 calculates a summarization score as the sum of the expression of selected markers, with the cell type-specific markers first nominated from expression profiles in purified bulk data or identified^{31,32} from single-cell data expression profiles and then prioritized according to their correlation with that cell type's proportion over synthetic admixtures (Supplemental Methods; https://github.com/sdomanskyi/mitten_TDC19; **Table S15**). Finally, Biogem, based on a previously published method,³³ uses robust linear modeling to perform deconvolution and differential expression-based feature selection to define the purified expression profiles (Supplemental Methods; https://github.com/giannimonaco/DREAMChallenge_Deconvolution; **Fig. S3**; **Table S16**).

2. Perspectives on deconvolution: As the field of genomics evolves, there emerges a key application of deconvolution which is about spatial transcriptomics. 10X Visium platform is a major one for measuring bulk transcriptomic profiles of cell aggregates on tissue slides and relies on deconvolution for interpretation. However, more fundamental deconvolution techniques like Seurat still dominate the field, primarily due to their user-friendliness. Benchmarking deconvolution methods could potentially spearhead progress in spatial transcriptomic analyses, fostering a deeper understanding of tissue microenvironments and pathological processes. I recommend that the authors acknowledge this trajectory and provide insights into how their study could contribute to this burgeoning area.

We agree that deconvolution of spatial transcriptomics (ST) data is an exciting and burgeoning field. As the Reviewer requests, we have added a perspective on this to Discussion:

Our focus in this study was on deconvolution of bulk RNA-seq, however spatial transcriptomics technologies are rapidly emerging. To date, many of these approaches rely on sequencing small populations of cells across tissue regions—for example the multicellular regions “spots” assayed by the Visium platform⁷ and the “pucks” of the Slide-seqV2 platform.⁶⁸ To accelerate development of ST algorithms, our purified expression profiles could be used to simulate data for assessment of ST deconvolution methods.

For the Reviewer’s convenience, we here include the above references:

7. Ståhl, P. L. *et al.* Visualization and analysis of gene expression in tissue sections by spatial transcriptomics. *Science* **353**, 78–82 (2016).
68. Stickels, R. R. *et al.* Highly sensitive spatial transcriptomics at near-cellular resolution with Slide-seqV2. *Nat. Biotechnol.* **39**, 313–319 (2021).

Additional comments:

1. This is a useful benchmarking study on a key topic of genomics - reference-based deconvolution of bulk transcriptomic data. However, it's a missed opportunity that established methods like MuSiC and prominent techniques in spatial transcriptomics (e.g., Seurat and cell2location) weren't factored into the comparative analysis. Including these could substantially enhance the impact of the paper.

We appreciate the Reviewer’s positive feedback that our study presents a useful benchmark for tumor deconvolution methods. We agree that spatial transcriptomics methods are becoming more common, however the considerations for their analysis are sufficiently different that they warrant a specialized evaluation, as we now describe in the Discussion:

ST deconvolution method development and assessment have considerations beyond that relevant in deconvolution of bulk RNA-seq data, including modeling of sparse (near single-cell) sequencing data, incorporating number of cells obtained from a matched histology image as a prior constraint, and sharing information across a spatial neighborhood. Others have reviewed⁶⁹ and benchmarked⁷⁰ methods in this field. We envision that the approaches outlined herein could be used in future studies as a blueprint to assess deconvolution algorithms tuned to ST data.

69. Zhang, Y. *et al.* Deconvolution algorithms for inference of the cell-type composition of the spatial transcriptome. *Comput. Struct. Biotechnol. J.* **21**, 176–184 (2023).
70. Li, H. *et al.* A comprehensive benchmarking with practical guidelines for cellular deconvolution of spatial transcriptomics. *Nat. Commun.* **14**, 1548 (2023).

2. Figure 1a's legend shouldn't use "boxes shaded blue" to refer to the coarse-grained cell types because the 8 blocks are not all rectangle boxes. Maybe "blocks" or "segments" rather than "boxes"?

We have completely revised **Fig 1**. In its revised legend, we simply refer generically to "shading" without specifying "box", "block", etc.

3. The diagram in Figure 1b (pertaining to the in vitro section) doesn't resonate intuitively. The text said that it's the RNAs that were mixed physically, but the diagram showed that cells were mixed and placed in culture dishes before sequencing, which seems incongruent with the actual experimental protocol and warrants reconsideration.

We thank the Reviewer for pointing this out. We have revised **Fig 1** (specifically panel **B**) so that it depicts mixing of RNA rather than cells.

Reviewer 3

Dear editor and authors,

In my opinion, the authors did address most of the raised points of concern very well. A benchmarking focused on both coarse and fine grained annotation is an added value to the public. The generated data is important for future tool development to allow for proper benchmarking against what is existing.

We are glad the Reviewer found that we addressed raised concerns well and that our study and data have value to the community.

Since I was not part of the initial submission, I still have some remarks.

Which of the methods hold proportional fold changes between samples? In some cases that might be more important than to predict the exact percentages themselves.

We agree with the reviewer that fold-change metrics can be a very informative approach to interpret biological data. We evaluated results computed during the Challenge using fold change and include them below.

We calculated fold change for the measured and predicted values and then calculated the Pearson correlation between the measured and predicted fold changes. More specifically, we sorted *both* the measured and predicted values in increasing order based on the *measured* values. We then computed the fold change between a sorted value (in the numerator) and its preceding sorted value (in the denominator). The results are shown below in **Fig. I**, which amends the fold change values to the original manuscript **Fig. 2**.

CIBERSORTx remains the best performer in the coarse-grained sub-Challenge based on fold change (**Fig. IA**). Nevertheless, the fold change correlations are universally poor. For example, no method achieves a fold change correlation greater than 0.5 in the coarse-grained sub-Challenge, whereas over half of the methods achieve a Pearson and Spearman correlation of raw values greater than 0.75.

Fig 1. Aggregate cross-sample performance of participant and comparator deconvolution methods. Aggregate score (primary metric: Pearson correlation; secondary metric: Spearman correlation; Pearson correlation over sorted fold change values) of participant (first submission only) and comparator methods in (A) coarse- and (B) fine-grained sub-Challenges over bootstraps ($n=1,000$; Methods). Comparator methods (bold) are shown only if their published reference signatures include all cell types in each respective sub-Challenge: CIBERSORTx (coarse-grained only) and xCell. Boxplots display median (center line), 25th and 75th percentiles (hinges), and 1.5x interquartile range (whiskers). Methods ordered by median Pearson correlation in respective sub-Challenge. DNN: deep neural network; ENS: ensemble; NMF: non-negative matrix factorization; NNLS: non-negative least squares; OTH: other; PI: probabilistic inference; REG: other regression; SUM: summary; SVR: support vector regression; UNK: unknown/unspecified; Frac: unnormalized fractions that need not sum to one; Norm: normalized scores (comparable across cell types and samples); Prop: proportions that sum to one.

We further investigated the cause of this poor performance by comparing raw (**Fig. IIA**) and fold change (**Fig. IIB**) correlations for CIBERSORTx predictions of B cells. Despite the strong correlation of *raw* predicted and measured B cell proportions ($R=0.96$; $p<2.2\times 10^{-16}$; **Fig. IIA**), the correlation was in the wrong direction and insignificant when the values were first converted to fold changes before correlating them ($R=-0.12$; $p=0.54$; **Fig. IIB**). Most measured fold changes are approximately one, whereas the few larger fold changes result from small-magnitude, raw measured values. This presents at least four complications for some or all methods: 1) The large fold changes, which have the most impact on the computed correlations, are evaluated at noisy, low magnitude values. Hence, the results are dominated by regimes we already understand to be difficult to predict. 2) Relatedly, despite the few outliers, the majority of fold changes are in a narrow range. A typical and more meaningful evaluation of fold change would use values spanning decades. 3) Methods were not trained to predict fold changes. Had they been, they would have implicitly weighted the low magnitude / high fold change values most

impacting the correlation. As such, evaluating them on metrics they were not developed to assess is unfair. This would remain true even if we designed new measured values spanning decades and incurred the additional time delay in evaluating them. 4) Some methods report (small) negative values. This is most likely to occur for the large fold change values, i.e., those dominating the correlation. It isn't clear how to calculate a fold change between a positive and a negative value.

While we appreciate the Reviewer's suggestion, we have opted not to include these new results for the above reasons.

Fig II. Comparison of correlation between raw and fold change values. B cell proportions predicted using CIBERSORTx were (A) compared directly to measured values or (B) first converted to fold changes and then compared to measured fold changes. R : Pearson correlation.

The paper might benefit from an overview figure, summarizing the metrics from the individual analysis and figure panels, so it is easier to compare the methods.

We appreciate this suggestion from the Reviewer. We have significantly revised **Fig 1** to clarify the different evaluations we performed. In particular, **Fig 1C** indicates that the methods were ranked based on cross-sample, within cell-type correlation. A new **Fig 1D** lists the three additional comparisons used in the post-competitive phase: 1) cross-cell type, within-sample correlation, 2) sensitivity (“spillover”), and 3) specificity (“limit of detection”). Additionally, the figure legend lists the subsequent figures that provide the corresponding results.

The created data will become available, which is good for future deconvolution tool releases to benchmark. It would be even better if the authors provide a script/tool to compare their own results with these in the DREAM challenge.

We agree that this would be of value to the community and would make our data most impactful. We have added a script and the following text in the Code Availability section to introduce it:

The script

<https://github.com/Sage-Bionetworks/Tumor-Deconvolution-Challenge/blob/master/analysis/validation-analysis/run-deconvolution-method-on-challenge-data.R> demonstrates how to run a deconvolution method against the Challenge data, using xCell as an example, and to compare it to the Challenge results.

I do have some additional minor remarks.

In the overview Figure 1 and from the result section (and even the methods) it is not fully clear to me to which level immune cells were in vitro purified?

We apologize that this was buried in the supplemental tables. We have made it explicit in a newly designed **Fig. 1B**, which we now refer to in text:

We isolated immune cells from healthy donors and obtained stromal, endothelial, and cancer cells from cell lines (**Fig. 1B** and **Tables S1** and **S2**; Methods).

In Figure 2: I would make the x-axis comparable for both coarse and fine grained annotation to ease interpretation.

To facilitate comparison across coarse- and fine-grained cell types, rounds, and metric (spearman vs pearson correlation), we have made the x axis (correlation) comparable across plots in **Figs 2, S4, and S5**, as requested by the Reviewer.

In multiple figures cell types are sorted based on alphabet. It would be handier for interpretation if T cell subtypes are all together instead of spread over the graph.

We appreciate that sorting by phenotype (e.g., such that CD8 subtypes are listed together) facilitates biological interpretation. We have made that change for **Fig S21**. Additionally, the purified populations in the spillover experiments are sorted by phenotype in **Figs 5A, S16, and S17**.

In other cases, cell types (and methods) are sorted by a quantitative measure. This facilitates answering questions such as what cell types were universally difficult to predict across methods? What method performed best in aggregate? etc. We have opted to keep this ordering as it is consistent with the primary objective quantification of the Challenge. Any quantitative ordering is now noted in the respective captions. In particular, in the heatmaps (**Figs 3A, S7, S9, S11, S14, S18, and S19**), boxplots (**Figs 3B, S6, S8, S10, S12, S13, S15, and S20**), and

dotplot (**Fig 7**), methods are ordered according to their mean Pearson correlation across cell types, and cell types are ordered according to their maximum Pearson correlation across methods. In **Fig 4** methods are ordered by median Pearson correlation across samples in respective sub-Challenge; in **Fig 5** methods and cell types are ordered according to their median spillover; in **Fig 6** methods are ordered according to their mean limit of detection.

It is not fully clear to me why CD4/CD8T cells are still a separate category while the subtypes are also included.

Not all of the baseline comparator methods are able to finely resolve subtypes of CD4/CD8 T cells. Nevertheless, since they are widely used and less granular resolution can still be useful, we opted to include both levels of representation of these cell types.

Reviewer 4

I co-reviewed this manuscript with one of the reviewers who provided the listed reports as part of the Nature Communications initiative to facilitate training in peer review and appropriate recognition for co-reviewers.

We appreciate Reviewer 4 co-reviewing our revised manuscript.

Reviewer 5

I co-reviewed this manuscript with one of the reviewers who provided the listed reports as part of the Nature Communications initiative to facilitate training in peer review and appropriate recognition for co-reviewers.

We similarly thank Reviewer 5 for co-reviewing our revised manuscript.

Reviewer #1 (Remarks to the Author):

The authors has addressed all my concerns.

Reviewer #2 (Remarks to the Author):

The authors have successfully addressed the majority of my previously raised concerns in their revised manuscript.

Acknowledging the structure of the teamwork, I recognize the difficulty in re-designing the benchmarking process and re-running the entire experiment. But it's worth noting that the widely cited MuSiC deconvolution method, with over 500 citations, was not included in the list of benchmarking algorithms. It may be beneficial to highlight this omission in the limitations section of the article.

Reviewer #2 (Remarks on code availability):

The Github link shows 404 error.

Reviewer #3 (Remarks to the Author):

Dear,

The authors did a large effort to react and perform additional analysis on the raised concerns. Especially the fold change analysis was highly appreciated. I am wondering if a RSME calculation would be a better metric compared to a correlation. I understand that the authors are not putting this in the manuscript, but I would like to see some of the thoughts mentioned in the discussion. The additional overview figure adds a lot of structure to the paper but the resolution appeared to be low. I am also not able to reach the GitHub file (run_deconvolution_method_...R).

I am happy with the changes and would just ask to double check the figure and table quality, as well as adding something about fold changes in the discussion session and make the code available.

Kind regards

Reviewer #3 (Remarks on code availability):

The code is not visible.

Point-by-point response

We appreciate the Nature Communications reviewers second round of reviews. We are pleased that they feel we have addressed the majority of their concerns. Doing so improved our manuscript considerably. We have made several additional changes to address further issues raised.

Reviewer 1

The authors has addressed all my concerns.

We are grateful for the Reviewer's previous comments. We are pleased they feel we have addressed their concerns.

Reviewer 2

The authors have successfully addressed the majority of my previously raised concerns in their revised manuscript.

We thank Reviewer 2 for their attention to our manuscript. We are glad they are satisfied with our response to their earlier concerns.

Acknowledging the structure of the teamwork, I recognize the difficulty in re-designing the benchmarking process and re-running the entire experiment. But it's worth noting that the widely cited MuSiC deconvolution method, with over 500 citations, was not included in the list of benchmarking algorithms. It may be beneficial to highlight this omission in the limitations section of the article.

We appreciate the Reviewer's flexibility in light of the Challenge structure. We have added an explicit reference to MuSiC and acknowledgement that this populare approach was not included in our benchmarking.

The Github link shows 404 error.

We apologize for our oversight in making the GitHub repository private. We have now made it public and provided versioned, openly accessible links to the code on GitHub (<https://github.com/Sage-Bionetworks/Tumor-Deconvolution-Challenge/releases/tag/v1.0.0>) and on Zenodo (<http://doi.org/10.5281/zenodo.11110924>), as noted in our Code Availability section.

Reviewer 3

The authors did a large effort to react and perform additional analysis on the raised concerns. Especially the fold change analysis was highly appreciated.

We are glad the Reviewer found that we addressed raised concerns.

I am wondering if a RSME calculation would be a better metric compared to a correlation. I understand that the authors are not putting this in the manuscript, but I would like to see some of the thoughts mentioned in the discussion.

In keeping with the standards at Nature Communications, we have now made all Source Data needed to create the figures available on Zenodo. This will allow others to compute additional metrics they find of interest. However, to keep the lengthy Discussion manageable, we have decided not to explicitly include consideration of RMSE versus correlation. RMSE is, however, included in our figures.

The additional overview figure adds a lot of structure to the paper but the resolution appeared to be low.

We appreciate Reviewer 3 asking for this overview figure. We agree that it helps organize our manuscript. We will work with the Nature Communications production team to ensure it is legible.

I am also not able to reach the GitHub file (run_deconvolution_method_...R).

We apologize that this was not accessible. As we noted above in response to a similar comment from Reviewer 2, our code is now publicly accessible on GitHub (<https://github.com/Sage-Bionetworks/Tumor-Deconvolution-Challenge/releases/tag/v1.0.0>) and Zenodo (<http://doi.org/10.5281/zenodo.11110924>).